# GLASU: A Communication-Efficient Algorithm for Federated Learning with Vertically Distributed Graph Data

## Abstract

Vertical federated learning (VFL) is a distributed learning paradigm, where computing clients collectively train a model based on the partial features of the same set of samples they possess. Current research on VFL focuses on the case when samples are independent, but it rarely addresses an emerging scenario when samples are interrelated through a graph. For graph-structured data, graph neural networks (GNNs) are competitive machine learning models, but a naive implementation in the VFL setting causes a significant communication overhead. Moreover, the analysis of the training is faced with a challenge caused by the biased stochastic gradients. In this paper, we propose a model splitting method that splits a backbone GNN across the clients and the server and a communication-efficient algorithm, GLASU, to train such a model. GLASU adopts lazy aggregation and stale updates to skip aggregation when evaluating the model and skip feature exchanges during training, greatly reducing communication. We offer a theoretical analysis and conduct extensive numerical experiments on real-world datasets, showing that the proposed algorithm effectively trains a GNN model, whose performance matches that of the backbone GNN when trained in a centralized manner.

## 1 Introduction

Vertical federated learning (VFL) is a newly developed machine learning scenario in distributed optimization, where clients share data with the same sample identity but each client possesses only a subset of the features for each sample. The goal is for the clients to collaboratively learn a model based on all features. Such a scenario appears in many applications, including healthcare, finance, and recommendation systems (Chen et al., 2020b; Liu et al., 2022). For example, in healthcare, each hospital may collect partial clinical data of a patient such that their conditions and treatments are best predicted through learning from the data collectively; in finance, banks or e-commerce providers may jointly analyze a customer's credit with their trade histories and personal information; and in recommendation systems, online social/review platforms may collect a user's comments and reviews left at different websites to predict suitable products for the user.

Most of the current VFL solutions (Chen et al., 2020b; Liu et al., 2022) treat the case where samples are independent, but omit their relational structure. However, the pairwise relationship between samples emerges in many occasions and it can be crucial in several learning scenarios, including the low-labeling-rate scenario in semi-supervised learning and the no-labeling scenario in self-supervised learning. Take the financial application as an example: customers and institutions are related through transactions. Such relations can be used to trace finance crimes such as money laundering, to assess the credit risk of a customer, and even to recommend products to them. Each bank and e-commerce provider can infer the relations of the financial individuals registered to them and create a relational graph, in addition to the individual customer information they possess.

One of the most effective machine learning models to handle relational data is graph neural networks (GNNs) (Kipf & Welling, 2016; Hamilton et al., 2017; Chen et al., 2018; Velickovic et al., 2018; Chen et al., 2020a). This model performs neighborhood aggregation in every feature transformation layer, such that the prediction of a graph node is based on not only the information of this node but also that of its neighbors. Although GNNs have been used in federated learning, a majority

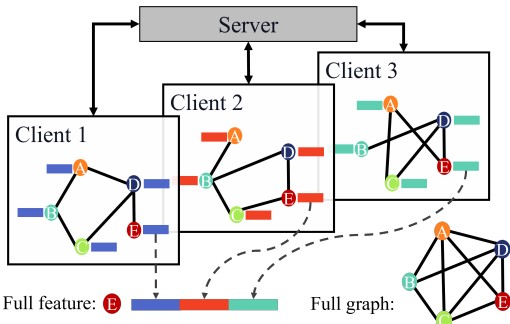

Figure 1: Data isolation of vertically distributed graph-structured data over three clients.

of the cases therein are horizontal: each client possesses a local dataset of graphs and all clients collaborate to train a unified model to predict graph properties, rather than node properties (He et al., 2021; Bayram & Rekik, 2021; Xie et al., 2021). Our case is different. We are concerned with *subgraph level, vertical* federated learning (Zhou et al., 2020; Ni et al., 2021): each client holds a subgraph of the global graph, part of the features for nodes in this subgraph, and part of the whole model; all clients collaboratively predict node properties. Our vertical setting is exemplified by not only the partitioning of node features, but also the (sub)graphs among the nodes.

The setting under our consideration is fundamentally challenging, because fully leveraging features within neighborhoods causes an enormous amount of communication. One method to design and train a GNN is that each client uses a local GNN to extract node representations from its own subgraph and the server aggregates these representations to make predictions (Zhou et al., 2020). The drawback of this method is that the partial features of a node outside one client's neighborhood are not used, even if this node appears in another client's neighborhood. Another method to train a GNN is to simulate centralized training: transformed features of each node are aggregated by the server, from where neighborhood aggregation is performed (Ni et al., 2021). This method suffers the communication overhead incurred in each layer computation.

In this work, we propose a federated GNN model and a communication-efficient training algorithm, named GLASU, for federated learning with vertically distributed graph data. The model is split across the clients and the server, such that the clients can use a majority of existing GNNs as the backbone, while the server contains no model parameters. The server only aggregates and disseminates computed data with the clients. The communication frequency between the clients and the server is mitigated through *lazy aggregation and stale updates* (hence the name of the method), with convergence guarantees. Moreover, GLASU can be considered as a framework that encompasses many well-known models and algorithms as special cases, including the work of Liu et al. (2022) when the subgraphs are absent, the work of Zhou et al. (2020) when all aggregations but the final one are skipped, the work of Ni et al. (2021) when no aggregations are skipped, and centralized training when only a single client exists.

We summarize the main contributions of this work below:

- Model design: We propose a flexible, federated GNN architecture that is compatible with a majority of existing GNN models.

- Algorithm design: We propose the communication-efficient GLASU algorithm to train the model. Therein, lazy aggregation saves communication for each joint inference round, through skipping some aggregation layers in the GNN; while stale updates further save communication by allowing the clients to use stale global information for multiple local model updates.

- Theoretical analysis: We provide theoretical convergence analysis for GLASU by addressing the challenges of biased stochastic gradient estimation caused by neighborhood sampling and correlated update steps caused by using stale global information.

- Numerical results: We conduct extensive experiments, together with ablation studies, to demonstrate that GLASU can achieve a comparable performance as the centralized model on multiple datasets and multiple GNN backbones, and that GLASU effectively saves communication.

## 1.1 PROBLEM SETUP

Consider $M$ clients, indexed by $m = 1, \ldots, M$, each of which holds a part of a graph with the node feature matrix $\mathbf{X} \in \mathbb{R}^{N \times d}$ and the edge set $\mathcal{E}$. Here, $N$ is the number of nodes in the graph and $d$ is the feature dimension. We assume that each client has the same node set and the same set of training labels, $\mathbf{y}$, but a different private edge set $\mathcal{E}_m$ and a non-overlapping node feature matrix $\mathbf{X}_m \in \mathbb{R}^{N \times d_m}$, such that $\mathcal{E} = \bigcup_{m=1}^{M} \mathcal{E}_m$, $\mathbf{X} = [\mathbf{X}_1, \ldots, \mathbf{X}_M]$, and $d = \sum_{m=1}^{M} d_m$. We denote the client dataset as $\mathcal{D}_m = \{\mathbf{X}_m, \mathcal{E}_m, \mathbf{y}\}$ and the full dataset as $\mathcal{D} = \{\mathbf{X}, \mathcal{E}, \mathbf{y}\}$. The task is for the clients to collaboratively infer the labels of nodes in the test set. See Figure 1 for an illustration.

## 1.2 GRAPH CONVOLUTIONAL NETWORK

The graph convolution network (GCN) (Kipf & Welling, 2016) is a typical example of the family of GNNs. Inside GCN, a graph convolution layer reads

$$\mathbf{H}[l+1] = \sigma\Big(\mathbf{A}(\mathcal{E}) \cdot \mathbf{H}[l] \cdot \mathbf{W}[l]\Big), \tag{1}$$

where $\sigma(\cdot)$ denotes the point-wise nonlinear activation function, $\mathbf{A}(\mathcal{E}) \in \mathbb{R}^{N \times N}$ denotes the adjacency matrix defined by the edge set $\mathcal{E}$ with proper normalization, $\mathbf{H}[l] \in \mathbb{R}^{N \times d[l]}$ denotes the node representation matrix at layer $l$, and $\mathbf{W}[l] \in \mathbb{R}^{d[l] \times d[l+1]}$ denotes the weight matrix at the same layer. The initial node representation matrix $\mathbf{H}[0] = \mathbf{X}$. The classifier is denoted as $\hat{\mathbf{y}} = f(\mathbf{H}[L], \mathbf{W}[L])$ with weight matrix $\mathbf{W}[L]$ and the loss function is denoted as $\ell(\mathbf{y}, \hat{\mathbf{y}})$. Therefore, the overall model parameter is $\mathbf{W} = \{\mathbf{W}[0], \ldots, \mathbf{W}[L-1], \mathbf{W}[L]\}$.

Mini-batch training of GCN (and GNNs in general) faces a scalability challenge, because to compute one or a few rows of $\mathbf{H}[L]$ (i.e., the representations of a mini-batch), it requires more and more rows of $\mathbf{H}[L-1], \mathbf{H}[L-2], \ldots$ recursively, in light of the multiplication with $\mathbf{A}(\mathcal{E})$ in (1). This is known as the *explosive neighborhood problem* unique to graph-structured data. Several sampling strategies were proposed in the past to mitigate the explosion; in this work we adopt the layer-wise sampling proposed by FastGCN (Chen et al., 2018). Starting from the output layer $L$, which is associated with a mini-batch of training nodes, $\mathcal{S}[L]$, we iterate over the layers backward such that at layer $l$, we sample a subset of neighbors for $\mathcal{S}[l+1]$, namely $\mathcal{S}[l]$. In doing so, at each layer we form a bipartite graph with edge set $\mathcal{E}(\mathcal{S}[l+1], \mathcal{S}[l]) = \{(i,j) | i \in \mathcal{S}[l+1], j \in \mathcal{S}[l]\}$. Then, each graph convolution layer becomes

$$\mathbf{H}[l+1][\mathcal{S}[l+1]] = \sigma\Big(\mathbf{A}(\mathcal{E}(\mathcal{S}[l+1], \mathcal{S}[l])) \cdot \mathbf{H}[l][\mathcal{S}[l]] \cdot \mathbf{W}[l]\Big), \tag{2}$$

where $\mathbf{A}(\mathcal{E}(\mathcal{S}[l+1], \mathcal{S}[l])) \in \mathbb{R}^{|\mathcal{S}[l+1]| \times |\mathcal{S}[l]|}$ is a properly scaled submatrix of $\mathbf{A}(\mathcal{E})$ and $\mathbf{H}[l][\mathcal{S}[l]]$ denotes the rows of $\mathbf{H}[l]$ corresponding to $\mathcal{S}[l]$. Such a mini-batch sampling and training procedure fundamentally differs from the usual mini-batch training for non-graph data in VFL.

## 1.3 RELATED WORKS

Federated learning on graph data generally fall under two categories, horizontal and vertical. The horizontal case can be considered graph-level, where each client possesses a collection of graphs and all clients collaborate to train a unified model to predict graph properties (Zhang et al., 2021; He et al., 2021; Bayram & Rekik, 2021; Xie et al., 2021). Applications include predicting molecular properties (Xie et al., 2021) and learning connectional brain templates (Bayram & Rekik, 2021). On the other hand, the vertical case can be considered subgraph-level, where each client holds a subgraph of the global graph, a part of the node features, and a part of the whole model (Zhou et al., 2020; Ni et al., 2021). The clients aim to collaboratively train a global model with the partial features and subgraphs to predict node properties (see Figure 1). Existing methods either fail to fully leverage the neighborhood information (Zhou et al., 2020) or incur expensive communication (Ni et al., 2021). Our approach addresses these shortcomings.

An additional scenario that does not fit into the above common categories is a node-level federated learning: the clients are connected by a graph and thus each of them is treated as a node. In other words, the clients, rather than the data, are graph-structured. For example, in Lalitha et al. (2019) and Meng et al. (2021), each client performs learning with its own data and they exchange data

through the communication graph; whereas in Caldarola et al. (2021) and Rizk & Sayed (2021), the server maintains the graph structure and uses a GNN to aggregate information (either models or data) collected from the clients.

## 2 PROPOSED APPROACH

In this section, we present the proposed model and the training algorithm GLASU for federated learning on vertically distributed graph data. The neighborhood aggregation in GNNs poses communication challenges distinct from conventional VFL. To mitigate this challenge, we propose lazy aggregation and stale updates to effectively reduce the communication between the clients and the server, while maintaining comparable prediction performance as centralized models. For notational simplicity, we present the approach by using the full-graph notation (1) but note that the implementation involves neighborhood sampling, where a more precise notation should follow (2), and that one can easily change the backbone from GCN to other GNN architectures.

### 2.1 GNN MODEL SPLITTING

We split the GNN model among the clients and the server, approximating a centralized model. Specifically, each GNN layer contains two sub-layers: the client GNN sub-layer and the server aggregation sub-layer. At the $l$-th layer, each client computes the local feature matrix

$$\mathbf{H}_m^+[l] = \sigma\Big(\mathbf{A}(\mathcal{E}_m) \cdot \mathbf{H}_m[l] \cdot \mathbf{W}_m[l]\Big)$$

with the local weight matrix $\mathbf{W}_m[l]$ and the local graph $\mathcal{E}_m$, where we use the superscript $^+$ to denote local representations before aggregation. Then, the server aggregates the clients' representations and outputs $\mathbf{H}[l+1]$ as

$$\mathbf{H}[l+1] = \mathrm{Agg}(\mathbf{H}_1^+[l], \ldots, \mathbf{H}_M^+[l]),$$

where $\mathrm{Agg}(\cdot)$ is an aggregation function. In this paper, we only consider parameter-free aggregations, examples of which include averaging $\mathrm{Avg}(\mathbf{H}_1^+[l], \ldots, \mathbf{H}_M^+[l]) = \frac{1}{M}\sum_{m=1}^M \mathbf{H}_m^+[l]$ and concatenation $\mathrm{Cat}(\mathbf{H}_1^+[l], \ldots, \mathbf{H}_M^+[l]) = [\mathbf{H}_1^+[l], \ldots, \mathbf{H}_M^+[l]]$. The server broadcasts the aggregated $\mathbf{H}[l+1]$ to the clients so that computation proceeds to the next layer. In the final layer, each client computes a prediction. This layer is the same among clients because they receive the same $\mathbf{H}[L]$.

The two aggregation operations of our choice have a few advantages.

- Parameter-free: Since the operations do not contain any learnable parameters, the server does not need to perform gradient computations.
- Memory-less: In the backward pass, these operations do not require data from the forward pass to back-propagate the gradients. For averaging, the server back-propagates $\frac{1}{M}\partial\mathbf{H}[l+1]$ to each client, while for concatenation, the server back-propagates the corresponding block of $\partial\mathbf{H}[l+1]$.
- Easy-to-implement: The server implementation is obviously easy because of the parameter-free and memory-less nature. Moreover, they enable parallelization and pipelining.

We illustrate in Figure 2 the split of one GNN layer among the clients and the server. Although our approach resembles federated split learning (SplitFed) (Thapa et al., 2022), there is a fine distinction. In SplitFed, each client can collaborate with the server to perform inference or model updates without accessing information from other clients; while in our case, all clients collectively perform the job. Our approach also differs from conventional VFL that splits the local feature processing and the final classifier among the clients and the server respectively, such that each model update requires a single U-shape communication (Chen et al., 2020b). In our case, due to the graph structure, each GNN layer contains one client-server interaction and the number of interactions is equal to the number of GNN layers (we will relax this in the following subsection).

Note that there are two types of aggregations in our model. One is the neighborhood aggregation (multiplying with matrix $\mathbf{A}(\mathcal{E}_m)$), as a signature of GNNs, that occurs in each client locally and incurs no communication between the clients and the server. The other one relates to the communication that happens when the server gathers the clients' partial representations and broadcasts back the aggregated representation.

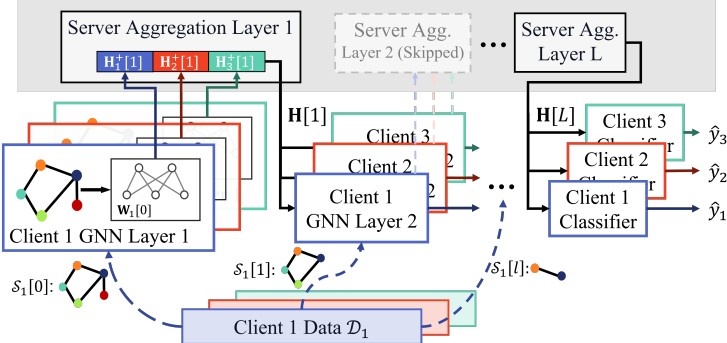

Figure 2: Illustration of the split model on $M = 3$ clients with lazy aggregation. In the model, the second server aggregation layer is skipped due to lazy aggregation, and the graph size used by each layer gradually decreases due to neighborhood sampling.

## 2.2 LAZY AGGREGATION

The development in the preceding subsection approximates a centralized model, but it is not communication friendly because each layer requires one round of client-server communication. We propose two communication-saving strategies in this subsection and the next. We first consider *lazy aggregation*, which skips aggregation in certain layers.

Instead of performing server aggregation at each layer, we specify a subset of $K$ indices, $\mathcal{I} = \{l_1, \ldots, l_K\} \subset [L]$, such that aggregation is performed only at these layers. That is, at a layer $l \in \mathcal{I}$, the server performs aggregation and broadcasts the aggregated representations to the clients, serving as the input to the next layer:

$$\mathbf{H}_m[l+1] = \mathbf{H}[l+1];$$

while at a layer $l \notin \mathcal{I}$, each client uses the local representations as the input to the next layer:

$$\mathbf{H}_m[l+1] = \mathbf{H}_m^+[l].$$

By doing so, the model skips the server aggregation sub-layer between $l_k$ and $l_{k+1}$, such that the amount of communication is reduced from $O(L)$ to $O(K)$.

There is a subtlety caused by neighborhood sampling: it requires additional rounds of communication. Neighborhood sampling is done similarly to FastGCN (see Section 1.2), but note that whenever server aggregation is performed, it must be done on the same set of sampled nodes across clients. Hence, the server takes the union of the clients' index sets $\mathcal{S}_m[l_k]$ and broadcasts $\mathcal{S}[l_k] = \bigcup_{m=1}^{M} \mathcal{S}_m[l_k]$ to the clients. On the other hand, when server aggregation is skipped at an layer $l \notin \mathcal{I}$, each client can use its own set of sampled nodes, $\mathcal{S}_m[l]$, that may differ across clients. Such a procedure is more flexible than conventional VFL, where strict sample synchronization is enforced. The sampling procedure is summarized in Algorithm 2 in the appendix.

## 2.3 STALE UPDATES

To further reduce communication, we consider *stale updates*, which skip aggregation in certain iterations and use stale node representations to perform model updates. The key idea is to fix the mini-batch, including the sampled neighbors at each layer, for training $Q$ iterations. In every other $Q$ iterations, the clients store the aggregated representations at the server aggregation layers. Then, in the subsequent iterations, every server aggregation is replaced by a local aggregation between a client's up-to-date node representations and other clients' stale node representations. By doing so, the clients and the server only need to communicate once in every $Q$ iterations.

Specifically, let a round of training contain $Q$ iterations and use $t$ to index the rounds. At the beginning of each round, the clients and the server jointly decide the set of nodes used for training at each layer. Then, they perform a joint inference on the representations $\mathbf{H}_m^{t,+}[l]$ at every layer $l \in \mathcal{I}$. Each client $m$ will store the "all but $m$" representation $\mathbf{H}_{-m}^t[l+1]$ through extracting such

---

**Algorithm 1** Training Procedure. All referenced algorithms are detailed in the appendix.

---

1: **for** $t = 0, \ldots, T - 1$ **do**
2:     **Server/Client**: Sample $\{\mathcal{S}_m^t[l]\}_{l=0}^L$ with Algorithm 2.
3:     **Client**: $\mathbf{W}_m^{t,0} = \begin{cases} \mathbf{W}_m^{t-1,Q}, & t > 0 \\ \mathbf{W}_m^0, & t = 0 \end{cases}$.
4:     **Server/Client**: $\{\mathbf{H}_{-m}^t[l+1]\}_{l \in \mathcal{I}} = \textbf{JointInference}(\mathbf{W}_m^{t,0}, \mathcal{D}_m, \{\mathcal{S}_m^t[l]\}_{l=0}^L)$. (Algorithm 3)
5:     **for** $q = 0, \ldots, Q - 1$ **do**
6:         **for** clients in parallel **do**
7:             $\mathbf{W}_m^{t,q+1} = \textbf{LocalUpdate}(\mathbf{W}_m^{t,q}, \mathcal{D}_m, \{\mathcal{S}_m^t[l]\}_{l=0}^L, \{\mathbf{H}_{-m}^t[l+1]\}_{l \in \mathcal{I}})$. (Algorithm 4)
8:         **end for**
9:     **end for**
10: **end for**
11: **Output:** $\{\mathbf{W}_m^{t,q}\}_{m=1}^M$

---

information from the aggregated representations $\mathbf{H}_m^t[l+1]$:

$$\mathbf{H}_{-m}^t[l+1] = \text{Extract}(\mathbf{H}_m^t[l+1], \mathbf{H}_m^{t,+}[l]).$$

For example, when the server aggregation is averaging, the extraction is

$$\text{Extract}(\mathbf{H}_m^t[l+1], \mathbf{H}_m^{t,+}[l]) = \mathbf{H}_m^t[l+1] - \frac{1}{M} \mathbf{H}_m^{t,+}[l].$$

Afterward, the clients perform $Q$ iterations of model updates, indexed by $q = 0, \ldots, Q - 1$, on the local parameters $\mathbf{W}_m^{t,q}$ in parallel, using the stored aggregated information $\mathbf{H}_{-m}^t[l+1]$ whenever a server aggregation is supposed to happen. In other words, the server aggregation is replaced by computation done locally, thus reducing a significant amount of communication. Because $\mathbf{H}_{-m}^t[l+1]$ is computed by using stale model parameters $\{\mathbf{W}_{m'}^{t,0}\}_{m' \neq m}$ at all iterations $q \neq 0$, this approach is called "stale updates."

The details are summarized in Algorithm 1, with subroutines given in Appendix A.

## 2.4 SPECIAL CASES

It is interesting to note that our model and the training algorithm encompass several well-known models and algorithms as special cases.

**Conventional VFL.** VFL algorithms can be viewed as a special case of the proposed algorithm, where $A(\mathcal{E}_m) = I$ for all $m$. In this case, no neighborhood sampling is needed and GLASU reduces to Liu et al. (2022).

**Existing VFL algorithms for graphs.** The model of Zhou et al. (2020) is a special case of our model, with $K = 1$; i.e., no communication is performed between the server and the clients except the final prediction layer. In this case, the clients omit the connections absent in the self subgraph but present in other clients' subgraphs. The model of Ni et al. (2021) is also a special case of our model, with $K = L$. This case requires communication at all layers and is less efficient.

**Centralized GNNs.** When there is a single client ($M = 1$), our setting is the same as centralized GNN training. Specifically, by letting $K = L$ and properly choosing the server aggregation function $\text{Agg}(\cdot)$, our split model can achieve the same performance as a centralized GNN model. Of course, using lazy aggregation ($K \neq L$) and choosing the server aggregation function as concatenation or averaging will make the split model different from a centralized GNN.

## 2.5 PRIVACY

Our training algorithm GLASU enables privacy protection because it is compatible with existing privacy preserving approaches, including secure aggregation (SA) and differential privacy (DP).

**SA** (Bonawitz et al., 2017; Hardy et al., 2017) is a form of secure multi-party computation approach used for aggregating information from a group of clients, without revealing the information of any

individual. This can be achieved by homomorphic encryption (Li et al., 2010; Hardy et al., 2017). In our case, when the server aggregation is averaging, homomorphic encryption can be directly applied.

**DP** (Wei et al., 2020) is a probabilistic protection approach. By injecting stochasticity to the local outputs, this approach guarantees that any attacker cannot distinguish the sample from the dataset up to a certain probability. DP can be applied either solely or in combination with SA to our algorithm in the server-client communication, to offer privacy protection on the client data.

## 3 CONVERGENCE ANALYSIS

With lazy aggregation and stale updates, GLASU is guaranteed to converge. To start the analysis, denote by $\mathcal{S}^t = \{\mathcal{S}_m^t[l]\}_{l=1,m=1}^{L,M}$ the samples used at round $t$ (which include all sampled nodes at different layers and clients); by $S = |\mathcal{S}_m^t[L]|$ the batch size; and by $\mathcal{L}(\mathbf{W}; \mathcal{S})$ the training objective, which is evaluated at the overall set of model parameters across clients, $\mathbf{W} = \{\mathbf{W}_m\}_{m=1}^M$, and a batch of samples, $\mathcal{S}$.

A few assumptions are needed (see Appendix B.1 for formal statements). **A1**: The loss function $\ell$ is $G_\ell$-smooth with $L_\ell$-Lipschitz gradient; and a client's prediction function $f_m$ is $G_f$-smooth with $L_f$-Lipschitz gradient. **A2**: The training objective $\mathcal{L}(\mathbf{W}; \mathcal{D})$ is bounded below by a finite constant $\mathcal{L}^\star$. **A3**: The samples $\mathcal{S}^t$ are sampled from $\mathcal{D}$ following Algorithm 2 in Appendix A.

For any round $t$ and iteration $q$ in the round, GLASU admits the following convergence guarantee.

**Theorem 1.** *Under assumptions A1–A3, by running Algorithm 1 with $\eta \leq \frac{1}{C_0 \cdot (1+2Q^2 M)}$, with probability at least $p = 1 - \delta$, the averaged gradient norm is bounded by:*

$$\frac{1}{TQ} \sum_{t=0}^{T-1} \sum_{Q=0}^{Q-1} \mathbb{E} \left\| \nabla \mathcal{L}(\mathbf{W}^{t,q}; \mathcal{D}) \right\|^2 \leq \frac{2(\mathcal{L}(\mathbf{W}^{0,0}) - \mathcal{L}^\star)}{\eta TQ} + \frac{28\eta M \cdot (C_0 + \sqrt{M+1}Q)}{3}\sigma, \quad (3)$$

*where $C_0 = G_\ell L_f + L_\ell G_f^2$ is constant and $\sigma > 0$ is a function of $\log(TQ/\delta), L_f, L_g, G_f$ and $G_g$.*

The detailed proofs are presented in Appendix B. Two key challenges in the analysis are: 1) the stochastic gradient estimation of the network is biased (i.e., $\mathbb{E}_{\mathcal{S}} \nabla \mathcal{L}(\mathbf{W}; \mathcal{S}) \neq \nabla \mathcal{L}(\mathbf{W}; \mathcal{D})$), even in centralized models; and 2) the stale updates in one communication round are correlated, as they are updated with the same samples. Hence, the general unbiasedness and independence assumptions on the stochastic gradients in the analysis of SGD-type algorithms do not apply. Instead, we follow the analysis in Ramezani et al. (2020) to bound the variance of the stochastic gradient in centralized GCN training, and extend the analysis in Liu et al. (2022) for VFL with correlated updates to our case with biased gradients.

## 4 NUMERICAL EXPERIMENTS

In this section, we conduct numerical experiments on a variety of datasets and demonstrate the effectiveness of GLASU in training with distributed graph data. We first compare the performance of GLASU with those tackling related settings under different assumptions on data distribution and communication. Then, we conduct ablation studies to show the equal criticality of the three components (GNN backbone, lazy aggregation, and stale updates) of GLASU. The experiments are conducted on a distributed cluster with three Tesla V100 GPUs communicated through Ethernet.

### 4.1 DATASETS

We use seven datasets (in three groups) with varying sizes and data distributions: the Planetoid collection (Yang et al., 2016), the HeriGraph collection (Bai et al., 2022), and the Reddit dataset (Hamilton et al., 2017). Each dataset in the HeriGraph collection (Suzhou, Venice, and Amsterdam) contains data readily distributed: three subgraphs and more than three feature blocks for each node. Hence, we use three clients, each of which handles one subgraph and one feature block. For the other four datasets (Cora, PubMed and CiteSeer in the Planetoid collection; and Reddit), each contains one single graph and thus we manually construct subgraphs through randomly sampling the edges and splitting the input features into non-overlapping blocks, so that each client handles one

| Dataset | # (Sub)graphs | # Nodes | # Edges | # Features | # Classes |
|---|---|---|---|---|---|
| Cora | 1 | $2,708$ | $10,556$ | $1,433$ | 7 |
| PubMed | 1 | $19,717$ | $88,648$ | $500$ | 3 |
| CiteSeer | 1 | $3,327$ | $9,104$ | $3,703$ | 6 |
| Suzhou | 3 | $3,137$ | $916,496$ | $979$ | 9 |
| Venice | 3 | $2,951$ | $534,513$ | $979$ | 9 |
| Amsterdam | 3 | $3,727$ | $1,271,171$ | $979$ | 9 |
| Reddit | 1 | $232,965$ | $114,615,892$ | $602$ | 41 |

Table 1: Dataset summary. For a dataset that contains a single graph, each of the $M$ clients holds a sampled subgraph from it. For the HeriGraph datasets, there are $M = 3$ clients, each of which holds a given subgraph.

| Dataset | Cent. (%) | StAl. (%) | Sim. (%) | GLASU-1 (%) | GLASU-4 (%) |
|---|---|---|---|---|---|
| Cora | $80.9 \pm 0.6$ | $74.6 \pm 0.5$ | $80.1 \pm 1.2$ | $81.0 \pm 1.3$ | $80.3 \pm 1.2$ |
| PubMed | $84.9 \pm 0.6$ | $77.2 \pm 0.5$ | $82.7 \pm 1.2$ | $82.3 \pm 1.6$ | $83.8 \pm 1.8$ |
| CiteSeer | $70.2 \pm 0.8$ | $64.4 \pm 0.5$ | $70.0 \pm 1.2$ | $70.0 \pm 1.7$ | $68.8 \pm 3.3$ |
| Suzhou | $94.3 \pm 0.3$ | $51.6 \pm 0.9$ | $93.5 \pm 0.6$ | $92.7 \pm 1.4$ | $90.4 \pm 0.8$ |
| Venice | $95.7 \pm 0.5$ | $33.5 \pm 2.1$ | $93.1 \pm 1.3$ | $92.2 \pm 0.6$ | $91.0 \pm 1.6$ |
| Amsterdam | $94.6 \pm 0.1$ | $59.8 \pm 1.0$ | $95.5 \pm 0.8$ | $93.1 \pm 0.8$ | $94.9 \pm 0.4$ |
| Reddit | $95.6 \pm 0.1$ | $87.3 \pm 0.3$ | $95.3 \pm 0.7$ | $95.7 \pm 0.6$ | $94.7 \pm 1.1$ |

Table 2: Test accuracy (%) on different datasets. The compared algorithms are Centralized training (Cent.), Standalone training (StAl.), Simulated centralized training (Sim.), GLASU with no stale updates, i.e., $Q = 1$ (GLASU-1), and GLASU with stale updates $Q = 4$ (GLASU-4).

subgraph and one feature block. The dataset statistics are summarized in Table 1 and more details are given in Appendix C.1.

## 4.2 RESULTS

We compare GLASU with three training methods: a) centralized training, where there is only a single client ($M = 1$), which holds the whole dataset without any data distribution and communication; b) standalone training, where each client trains a model with its local data only and they do not communicate; c) simulated centralized training (Ni et al., 2021), where each client possesses the full graph but only the partial features, so that it simulates centralized training through server aggregation in each GNN layer. None of these compared methods fits VFL but they offer good references for understanding the performance of VFL on graph data. Except for centralized training, the number of clients $M = 3$. The number of training rounds, $T$, and the learning rate $\eta$ are optimized through grid search. See Appendix C.2 for details.

We use GCNII (Chen et al., 2020a) as the backbone GNN. One layer of GCNII reads

$$\mathbf{H}[l + 1] = \sigma\Big(\big((1 - \alpha[l])\mathbf{A}(\mathcal{E})\mathbf{H}[l] + \alpha[l]\mathbf{H}[0]\big)\big((1 - \beta[l])I + \beta[l]\mathbf{W}[l]\big)\Big),$$

which effectively includes two residual connections. This backbone reduces over-smoothing and results in better prediction accuracy than GCN. We set the number of layers $L = 4$ and the mini-batch size $S = 16$. For neighborhood sampling, we sample three neighbors per node in $\mathcal{S}[l + 1]$ and take the union of the sampled neighbors to form $\mathcal{S}[l]$. For lazy aggregation, we set $K = 2$.

Table 2 reports the classification accuracy of GLASU and the compared training methods, after five runs. As expected, standalone training produces the worst results, because each client uses only local information and misses edges and node features present in other clients. The centralized training and its simulated version lead to similar performance, also as expected. Our method GLASU is quite comparable with these two methods. Using stale updates ($Q = 4$) is generally outperformed by no stale updates, but occasionally it is better (see PubMed and Amsterdam). The gain in using stale updates occurs in timing, as will be demonstrated in the ablation study next.

### 4.3 ABLATION STUDY

To further investigate how each component of the proposed approach affects the performance, we conduct an ablation study on a) the backbone GNN model, b) the lazy aggregation parameter $K$, and c) the stale update parameter $Q$. The experiments use PubMed and Planetoid for illustration.

**Backbone model:** We compare the performance of GLASU on three backbone models: GCN, GAT (Velickovic et al., 2018), and GCNII. The learning rate for each backbonoe is tuned to their best performance. The test accuracy on PubMed is shown in Figure 3. We see that GLASU can take different GNNs as the backbone and reach a similar prediction performance.

**Lazy aggregation:** We investigate the performance of GLASU with different numbers of aggregation layers. We use a 4-layer GCNII as the backbone and set $K = 1, 2, 4$. The test accuracy and the runtime are listed in Table 3. We observe that the runtime decreases drastically when using fewer and fewer aggregation layers; from $K = 4$ to $K = 1$, the reduction in runtime is $37.4\%$ for PubMed and $58.2\%$ for Amsterdam. Meanwhile, there appears to be a sweet spot in terms of accuracy: $K = 2$ performs the best.

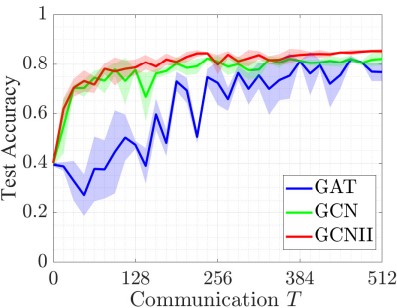

Figure 3: Test accuracy with different backbone GNNs on PubMed.

| PubMed | | | |
|---|---|---|---|
| # Layer | $K = 4$ | $K = 2$ | $K = 1$ |
| Accuracy (%) | $82.5 \pm 1.0$ | $83.8 \pm 1.8$ | $82.2 \pm 0.7$ |
| Runtime ($s$) | $130 \pm 12$ | $96.6 \pm 9.9$ | $81.3 \pm 6.5$ |
| Amsterdam | | | |
| # Layer | $K = 4$ | $K = 2$ | $K = 1$ |
| Accuracy (%) | $93.6 \pm 0.7$ | $94.9 \pm 0.4$ | $92.0 \pm 1.7$ |
| Runtime ($s$) | $913 \pm 76$ | $544 \pm 44$ | $382 \pm 35$ |

Table 3: Test accuracy and runtime of GLASU with different number of lazy aggregation layers $K = 4, 2, 1$ on PubMed and Amsterdam.

**Stale updates:** To investigate the time saving due to the use of stale updates, we experiment with a few choices of $Q$: 2, 4, 8, and 16. We report the time to reach the same test accuracy in Table 4. We see that stale updates help speed up training by using fewer communication rounds; this trend occurs on the Amsterdam dataset even when taking $Q$ as large as 16. The trend is also noticeable on PubMed, but at some point ($Q = 8$) it is reverted, likely because it gets harder and harder to reach the desired prediction accuracy. We speculate that the target 82% can never be achieved at $Q = 16$.

| | # Stale Update | $Q = 2$ | $Q = 4$ | $Q = 8$ | $Q = 16$ |
|---|---|---|---|---|---|
| PubMed | Accuracy (%) | $82.5 \pm 1.6$ | $82.0 \pm 2.4$ | $82.1 \pm 0.3$ | N/A |
| | Runtime ($s$) | $66.1 \pm 5.0$ | $43.8 \pm 4.0$ | $88.9 \pm 7.4$ | $> 128$ |
| Amsterdam | Accuracy (%) | $89.2 \pm 0.4$ | $89.3 \pm 0.7$ | $90.7 \pm 0.5$ | $90.3 \pm 1.1$ |
| | Runtime ($s$) | $1323 \pm 44$ | $521 \pm 44$ | $324 \pm 31$ | $250 \pm 24$ |

Table 4: Runtime of GLASU with different number of stale updates $Q = 2, 4, 6, 16$, **when reaching** 82% **test accuracy on PubMed and** 89% **on Amsterdam**.

## 5 CONCLUSION

We have presented a flexible model splitting approach for VFL with vertically distributed graph data and proposed a communication-efficient algorithm, GLASU, to train the resulting GNN. Due to the graph structure among the samples, VFL on GNNs incurs heavy communication and poses an extra challenge in the convergence analysis, as the stochastic gradients are no longer unbiased. To overcome these challenges, our approach uses lazy aggregation to skip server-client communication and stale global information to update local models, leading to significant communication reduction. Moreover, our analysis makes no assumptions on unbiased gradients. We provide extensive experiments to show the flexibility of the model and the communication saving in the training.

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

## A    SUBROUTINES IN ALGORITHM 1

---
**Algorithm 2** Sampling Procedure
---

**Client:**
**for** $k = K, \dots, 2$ **do**
   Receive($\mathcal{S}[l_k]$).
   Set $\mathcal{S}_m[l_k] = \mathcal{S}[l_k]$.
   **for** $l = l_k - 1, \dots, l_{k-1}$ **do**
      Uniformly randomly sample indices $\mathcal{S}_m[l]$
      from neighbors of $\mathcal{S}_m[l + 1]$.
   **end for**
   Send($\mathcal{S}_m[l_{k-1}]$) **if** $k > 2$.
**end for**
**Output:** $\{\mathcal{S}_m[l]\}_{l=0}^{L}$

**Server:**
Uniformly and independently sample
   indices $\mathcal{S}[L]$ from training set.
Broadcast($\mathcal{S}[L]$).
**for** $k = K - 1, \dots, 2$ **do**
   Aggregate($\mathcal{S}_m[l_k]$).
   Compute $\mathcal{S}[l_k] = \bigcup_{m=1}^{M} \mathcal{S}_m[l_k]$.
   Broadcast($\mathcal{S}[l_k]$).
**end for**

---
**Algorithm 3** JointInference
---

**Client:**
**Input:** $\mathbf{W}_m, \mathcal{D}_m, \{\mathcal{S}_m[l]\}_{l=0}^{L}$
Set $\mathbf{H}_m[0] = \mathbf{X}_m[\mathcal{S}_m[0]]$
**for** $l = 0, \dots, L - 1$ **do**
   $\mathbf{H}_m^{+}[l] = \sigma(\mathbf{A}(\mathcal{E}(\mathcal{S}[l+1], \mathcal{S}[l]))\mathbf{H}_m[l]\mathbf{W}_m[l])$
   **if** $l \in \mathcal{I}$ **then**
      Send $\mathbf{H}_m^{+}[l]$ to server
      Receive $\mathbf{H}_m[l + 1]$
      $\mathbf{H}_{-m}[l + 1] = \text{Extract}(\mathbf{H}_m[l + 1], \mathbf{H}_m^{+}[l])$
   **else**
      Set $\mathbf{H}_m[l + 1] = \mathbf{H}_m^{+}[l]$
   **end if**
**end for**
**Output:** $\{\mathbf{H}_{-m}[l + 1]\}_{l \in \mathcal{I}}$

**Server:**
**for** $l \in \mathcal{I}$ **do**
   $\mathbf{H}[l + 1] = \text{Agg}(\mathbf{H}_1^{+}[l], \dots, \mathbf{H}_M^{+}[l])$.
   Broadcast $\mathbf{H}[l + 1]$.
**end for**

---
**Algorithm 4** LocalUpdate
---

**Input:** $\mathbf{W}_m^{t,q}, \mathcal{D}_m, \{\mathcal{S}_m^{t}[l]\}_{l=0}^{L}, \{\mathbf{H}_{-m}^{t}[l+1]\}_{l \in \mathcal{I}}$
Set $\mathbf{H}_m^{t,q}[0] = \mathbf{X}_m[\mathcal{S}_m^{t}[0]]$
**for** $l = 0, \dots, L - 1$ **do**
   $\mathbf{H}_m^{t,q,+}[l] = \sigma(\mathbf{A}(\mathcal{E}(\mathcal{S}_m^{t}[l+1], \mathcal{S}_m^{t}[l]))\mathbf{H}_m^{t,q}[l]\mathbf{W}_m^{t,q}[l])$
   **if** $l \in \mathcal{I}$ **then**
      Set $\mathbf{H}_m^{t,q}[l + 1] = \text{Agg}(\mathbf{H}_{-m}^{t}[l + 1], \mathbf{H}_m^{t,q,+}[l])$
   **else**
      Set $\mathbf{H}_m^{t,q}[l + 1] = \mathbf{H}_m^{t,q,+}[l]$
   **end if**
**end for**
Compute loss $\mathcal{L}_m^{t,q} = \ell\left(\mathbf{y}[\mathcal{S}_m^{t}[L]], f_m(\mathbf{H}_m^{t,q}[L], \mathbf{W}_m[L])\right)$
**Output:** $\mathbf{W}_m^{t,q+1} = \mathbf{W}_m^{t,q} - \eta^{t,q} \nabla_{\mathbf{W}_m^{t,q}} \mathcal{L}_m^{t,q}$

---

## B    PROOFS FOR SECTION 3

### B.1    ASSUMPTIONS

**Assumption 1** (Smooth function and Lipschitz gradient). *The loss function $\ell$ is $G_\ell$-smooth with $L_\ell$-Lipschitz gradient, i.e.,*

$$\|\ell(\mathbf{y}, \mathcal{S}, \mathbf{W}) - \ell(\mathbf{y}, \mathcal{S}, \mathbf{W}')\| \le G_\ell \|\mathbf{W} - \mathbf{W}'\|$$

$$\|\nabla_{\mathbf{W}} \ell(\mathbf{y}, \mathcal{S}, \mathbf{W}) - \nabla_{\mathbf{W}'} \ell(\mathbf{y}, \mathcal{S}, \mathbf{W}')\| \le L_\ell \|\mathbf{W} - \mathbf{W}'\|, \quad \forall\, \mathbf{W}, \mathbf{W}'$$

and each client's prediction function $f_m$ is $G_f$-smooth with $L_f$-Lipschitz gradient, i.e.,

$$\|f_m(\mathcal{S}, \mathbf{W}_m) - f_m(\mathcal{S}, \mathbf{W}'_m)\| \leq G_f \|\mathbf{W}_m - \mathbf{W}'_m\|$$

$$\left\|\nabla_{\mathbf{W}_m} f_m(\mathcal{S}, \mathbf{W}_m) - \nabla_{\mathbf{W}'_m} f_m(\mathcal{S}, \mathbf{W}_m)\right\| \leq L_f \|\mathbf{W}_m - \mathbf{W}'_m\|, \quad \forall\, \mathbf{W}_m, \mathbf{W}'_m, \forall\, m.$$

**Assumption 2** (Lower-bounded function). *The training objective is bounded below; that is, there exists a constant $\mathcal{L}^\star > -\infty$ such that for all $\{\mathbf{W}_m\}$, it satisfies that*

$$\mathcal{L}(\{\mathbf{W}_m\}) \geq \mathcal{L}^\star.$$

**Assumption 3** (Uniform sampling). *At each iteration $t$, the server and the clients uniformly sample nodes $\{\mathcal{S}_m[l]\}_{l=0}^L$, with $|\mathcal{S}[L]| = S$, according to Algorithm 2.*

### B.2 PROOF OF THEOREM 1

We first note the following useful relation:

$$\|a + b\|^2 = \|a - c + c - b\|^2 \leq (1 + \alpha)\|a - c\|^2 + (1 + \frac{1}{\alpha})\|c - b\|^2, \quad \forall \alpha > 0. \tag{4}$$

For notation simplicity, let us denote the expectation conditioned on all the information before iteration $t$ as

$$\mathbb{E}^t[\,\cdot\,] = \mathbb{E}_{\mathcal{S}^t}[\,\cdot\,|\mathbf{W}^{t-1,Q}, \ldots, \mathbf{W}^{0,0}, \mathcal{S}^{t-1}, \ldots, \mathcal{S}^0];$$

denote the "all-but-$m$" vector as $(\cdot)_{-m}$, (e.g., the collection of all client parameters except for client $m$ is $\mathbf{W}_{-m} = \{\mathbf{W}_{m'}\}_{m' \neq m}$); denote the client model updated with data $\mathcal{S}$ as $\mathbf{W}_m(\mathcal{S})$; denote the gradient evaluated with data $\mathcal{S}$ on parameter $\mathbf{W}_m$ as $\nabla\mathcal{L}(\mathbf{W}_m(\mathcal{S}), \mathcal{S})$; and denote the stacked gradient of all clients as $\mathbf{G} = [\nabla\mathcal{L}(\mathbf{W}_1(\mathcal{S}), \mathcal{S}), \ldots, \nabla\mathcal{L}(\mathbf{W}_M(\mathcal{S}), \mathcal{S})]$. Then, the update rule can be rewritten as:

$$\mathbf{W}^{t,q+1}(\mathcal{S}^t) = \mathbf{W}^{t,q}(\mathcal{S}^t) - \eta\mathbf{G}^{t,q}. \tag{5}$$

In addition, let us define a virtual model sequence updated with *full data* as $\mathbf{W}(\mathcal{D})$, i.e.,

$$\mathbf{W}^{t,q+1}(\mathcal{D}) = \mathbf{W}^{t,q}(\mathcal{D}) - \eta\nabla\mathcal{L}(\mathbf{W}^{t,q}(\mathcal{D}), \mathcal{D}). \tag{6}$$

We can bound the variance of the stochastic gradient at any round $t$ and iteration $q = 0$ with the following lemma:

**Lemma 1** (Bounded variance). *Under Assumptions 1–3, with probability at least $p = 1 - \delta$, the variance of the stochastic gradient is bounded by:*

$$\mathbb{E}^t\left[\left\|\nabla\mathcal{L}(\mathbf{W}; \mathcal{S}^t) - \nabla\mathcal{L}(\mathbf{W}; \mathcal{D})\right\|^2\right] \leq \sigma, \quad \forall\, \mathbf{W} \text{ independent of } \mathcal{S}^t,$$

$$\text{where } \sigma = 64 G_\ell^2 L_f^2 \log\left(\frac{2d}{\delta}\right) + 128 L_\ell^2 \left(G_f^4 + \frac{1}{S}\right)\left(\log\left(\frac{2d}{\delta}\right) + \frac{1}{4}\right). \tag{7}$$

The main technique for proving this lemma is to use the matrix Bernstein inequality (Tropp, 2015) to bound the variance of the stochastic gradients and the variance of the expectation for each client. The proof steps of Lemma 1 follows the same steps in the proofs for Lemmas 5 and 6 of Ramezani et al. (2020), so we omit them here.

Further, we bound the Lipschitz constant of the total loss function in the following lemma:

**Lemma 2** (Lipschitz gradient). *Under Assumptions 1–3, the full gradient and each partial gradient of the objective $\mathcal{L}(\mathbf{W}, \mathcal{S})$ are Lipschitz continuous with uniform constant $C_0 = G_\ell L_f + G_f^2 L_\ell$,*

$$\|\nabla_{\mathbf{W}}\mathcal{L}(\mathbf{W}, \mathcal{S}) - \nabla_{\mathbf{W}'}\mathcal{L}(\mathbf{W}', \mathcal{S})\| \leq C_0 \|\mathbf{W} - \mathbf{W}'\|, \quad \forall\, \mathbf{W}, \mathbf{W}'$$

$$\left\|\nabla_{\mathbf{W}_m}\mathcal{L}(\mathbf{W}, \mathcal{S}) - \nabla_{\mathbf{W}'_m}\mathcal{L}(\mathbf{W}', \mathcal{S})\right\| \leq C_0 \|\mathbf{W} - \mathbf{W}'\|, \quad \forall\, \mathbf{W}, \mathbf{W}', \forall\, m.$$

The proof of Lemma 2 is given below in Section B.3.

With the above results, we begin our proof for Theorem 1. First, applying Lemma 2, we have:

$$\mathcal{L}(\mathbf{W}^{t,q+1}, \mathcal{D}) - \mathcal{L}(\mathbf{W}^{t,q}, \mathcal{D}) \leq \left\langle\nabla\mathcal{L}(\mathbf{W}^{t,q}, \mathcal{D}), \mathbf{W}^{t,q+1} - \mathbf{W}^{t,q}\right\rangle + \frac{C_0}{2}\left\|\mathbf{W}^{t,q+1} - \mathbf{W}^{t,q}\right\|^2$$

$$\overset{(a)}{=} -\eta\left\langle\nabla\mathcal{L}(\mathbf{W}^{t,q}, \mathcal{D}), \mathbf{G}^{t,q}\right\rangle + \frac{C_0\eta^2}{2}\left\|\mathbf{G}^{t,q}\right\|^2$$

$$\overset{(b)}{=} -\frac{\eta}{2}\left(\left\|\nabla\mathcal{L}(\mathbf{W}^{t,q}, \mathcal{D})\right\|^2 + \left\|\mathbf{G}^{t,q}\right\|^2 - \left\|\nabla\mathcal{L}(\mathbf{W}^{t,q}, \mathcal{D}) - \mathbf{G}^{t,q}\right\|^2\right) + \frac{C_0\eta^2}{2}\left\|\mathbf{G}^{t,q}\right\|^2$$

$$= -\frac{\eta}{2}\left\|\nabla\mathcal{L}(\mathbf{W}^{t,q}, \mathcal{D})\right\|^2 - \frac{\eta}{2}(1 - \eta C_0)\left\|\mathbf{G}^{t,q}\right\|^2 + \frac{\eta}{2}\left\|\nabla\mathcal{L}(\mathbf{W}^{t,q}, \mathcal{D}) - \mathbf{G}^{t,q}\right\|^2, \tag{8}$$

where step $(a)$ applies the update rule of Algorithm 4 and step $(b)$ uses the fact that $\langle a, b \rangle = \frac{1}{2}\left(\|a\|^2 + \|b\|^2 - \|a-b\|^2\right)$. Taking expectation, we have:

$$
\begin{aligned}
\mathbb{E}^t[\mathcal{L}(\mathbf{W}^{t,q+1}, \mathcal{D}) - \mathcal{L}(\mathbf{W}^{t,q}, \mathcal{D})] &\leq -\frac{\eta}{2}\,\mathbb{E}^t\left\|\nabla\mathcal{L}(\mathbf{W}^{t,q}, \mathcal{D})\right\|^2 \\
&\quad - \frac{\eta}{2}(1 - \eta C_0)\,\mathbb{E}^t\left\|\mathbf{G}^{t,q}\right\|^2 + \frac{\eta}{2}\,\mathbb{E}^t\left\|\nabla\mathcal{L}(\mathbf{W}^{t,q}, \mathcal{D}) - \mathbf{G}^{t,q}\right\|^2 \\
&\stackrel{(a)}{=} -\frac{\eta}{2}\,\mathbb{E}^t\left\|\nabla\mathcal{L}(\mathbf{W}^{t,q}, \mathcal{D})\right\|^2 + \frac{\eta}{2}\,\mathbb{E}^t\left\|\nabla\mathcal{L}(\mathbf{W}^{t,q}, \mathcal{D}) - \mathbf{G}^{t,q}\right\|^2 \\
&\quad - \frac{\eta}{2}(1 - \eta C_0)(\left\|\mathbb{E}^t\,\mathbf{G}^{t,q}\right\|^2 + \mathbb{E}^t\left\|\mathbf{G}^{t,q} - \mathbb{E}^t\,\mathbf{G}^{t,q}\right\|^2) \\
&\stackrel{(b)}{\leq} -\frac{\eta}{2}\,\mathbb{E}^t\left\|\nabla\mathcal{L}(\mathbf{W}^{t,q}, \mathcal{D})\right\|^2 - \frac{\eta}{2}(1 - \eta C_0)(\left\|\mathbb{E}^t\,\mathbf{G}^{t,q}\right\|^2 + \mathbb{E}^t\left\|\mathbf{G}^{t,q} - \mathbb{E}^t\,\mathbf{G}^{t,q}\right\|^2) \\
&\quad + \frac{\eta}{2}\left((1 + \frac{1}{\eta C_0})\,\mathbb{E}^t\left\|\nabla\mathcal{L}(\mathbf{W}^{t,q}, \mathcal{D}) - \mathbb{E}^t\,\mathbf{G}^{t,q}\right\|^2 + (1 + \eta C_0)\,\mathbb{E}^t\left\|\mathbb{E}^t\,\mathbf{G}^{t,q} - \mathbf{G}^{t,q}\right\|^2\right) \\
&= -\frac{\eta}{2}\,\mathbb{E}^t\left\|\nabla\mathcal{L}(\mathbf{W}^{t,q}, \mathcal{D})\right\|^2 - \frac{\eta}{2}(1 - \eta C_0)\left\|\mathbb{E}^t\,\mathbf{G}^{t,q}\right\|^2 + \eta^2 C_0 \underbrace{\mathbb{E}^t\left\|\mathbf{G}^{t,q} - \mathbb{E}^t\,\mathbf{G}^{t,q}\right\|^2}_{\text{Term 1}} \\
&\quad + \frac{1 + \eta C_0}{2 C_0}\underbrace{\mathbb{E}^t\left\|\nabla\mathcal{L}(\mathbf{W}^{t,q}, \mathcal{D}) - \mathbb{E}^t\,\mathbf{G}^{t,q}\right\|^2}_{\text{Term 2}},
\end{aligned}
\tag{9}
$$

where step $(a)$ uses the fact that $\mathbb{E}(X)^2 = \mathbb{E}(X^2) + \mathbb{E}(X - \mathbb{E}(X))^2$ and step $(b)$ uses (4) with $\alpha = \eta C_0$. Next, we bound Term 1 and Term 2 in the above inequality separately.

### B.2.1 BOUND OF TERM 1

First, we can rewrite $\mathbb{E}^t[\left\|\mathbf{G}^{t,q} - \mathbb{E}^t\,\mathbf{G}^{t,q}\right\|^2]$ as:

$$
\begin{aligned}
\mathbb{E}^t[\left\|\mathbf{G}^{t,q} - \mathbb{E}^t\,\mathbf{G}^{t,q}\right\|^2] &= \sum_{m=1}^{M}\mathbb{E}^t\left[\left\|\nabla\mathcal{L}(\mathbf{W}_m^{t,q}(\mathcal{S}^t), \mathcal{S}^t) - \mathbb{E}_{\mathcal{S}}\nabla\mathcal{L}(\mathbf{W}_m^{t,q}(\mathcal{S}), \mathcal{S})\right\|^2\right] \\
&\stackrel{(a)}{\leq} \sum_{m=1}^{M}\underbrace{\mathbb{E}^t\left[\left\|\nabla\mathcal{L}(\mathbf{W}_m^{t,q}(\mathcal{S}^t), \mathcal{S}^t) - \nabla\mathcal{L}(\mathbf{W}_m^{t,q}(\mathcal{D}), \mathcal{D})\right\|^2\right]}_{\triangleq A_m^{t,q}},
\end{aligned}
\tag{10}
$$

where step $(a)$ uses the fact that $\mathbb{E}(X - \mathbb{E}(X))^2 \leq \mathbb{E}(X - Y)^2$ for all constant $Y$. Then, we can bound $A_m^{t,q}$ as follows. When $q = 0$, by Lemma 1, we obtain that $A_m^{t,0} \leq \sigma$ holds with probability $1 - \delta$. In general, when $q \geq 1$, we have:

$$
\begin{aligned}
A_m^{t,q} &\stackrel{(4)}{\leq} 2\,\mathbb{E}^t\left[\left\|\nabla\mathcal{L}(\mathbf{W}_m^{t,q}(\mathcal{S}^t), \mathcal{S}^t) - \nabla\mathcal{L}(\mathbf{W}_m^{t,q}(\mathcal{D}), \mathcal{S}^t)\right\|^2\right] \\
&\quad + 2\,\mathbb{E}^t\left[\left\|\nabla\mathcal{L}(\mathbf{W}_m^{t,q}(\mathcal{D}), \mathcal{S}^t) - \nabla\mathcal{L}(\mathbf{W}_m^{t,q}(\mathcal{D}), \mathcal{D})\right\|^2\right] \\
&\stackrel{(a)}{\leq} 2 C_0^2\,\mathbb{E}^t\left[\left\|\mathbf{W}_m^{t,q}(\mathcal{S}^t) - \mathbf{W}_m^{t,q}(\mathcal{D})\right\|^2\right] + 2\,\mathbb{E}^t\left[\left\|\nabla\mathcal{L}(\mathbf{W}_m^{t,q}(\mathcal{D}), \mathcal{S}^t) - \nabla\mathcal{L}(\mathbf{W}_m^{t,q}(\mathcal{D}), \mathcal{D})\right\|^2\right] \\
&\stackrel{(b)}{\leq} 2 C_0^2\,\mathbb{E}^t\left[\left\|\mathbf{W}_m^{t,q}(\mathcal{S}^t) - \mathbf{W}_m^{t,q}(\mathcal{D})\right\|^2\right] + 2\sigma,
\end{aligned}
\tag{11}
$$

which holds with probability $1 - \delta$. Here, step $(a)$ applies Lemma 2 to the first term and step $(b)$ applies Lemma 1 to the second term. Then, we bound $\mathbb{E}^t\left[\left\|\mathbf{W}_m^{t,q}(\mathcal{S}^t) - \mathbf{W}_m^{t,q}(\mathcal{D})\right\|^2\right]$ in the above equation as:

$$
\begin{aligned}
&\mathbb{E}^t\left[\left\|\mathbf{W}_m^{t,q}(\mathcal{S}^t) - \mathbf{W}_m^{t,q}(\mathcal{D})\right\|^2\right] \\
&\stackrel{(a)}{=} \mathbb{E}^t\left[\left\|\mathbf{W}_m^{t,0} - \eta\sum_{q'=0}^{q-1}\nabla\mathcal{L}(\mathbf{W}_m^{t,q'}(\mathcal{S}^t), \mathcal{S}^t) - \left(\mathbf{W}_m^{t,0} - \eta\sum_{q'=0}^{q-1}\nabla\mathcal{L}(\mathbf{W}_m^{t,q'}(\mathcal{D}), \mathcal{D})\right)\right\|^2\right]
\end{aligned}
$$

$$
\overset{(b)}{=} \eta^2 \, \mathbb{E}^t \left[ \left\| \sum_{q'=0}^{q-1} \left( \nabla \mathcal{L}(\mathbf{W}_m^{t,q'}(\mathcal{S}^t), \mathcal{S}^t) - \nabla \mathcal{L}(\mathbf{W}_m^{t,q'}(\mathcal{D}), \mathcal{D}) \right) \right\|^2 \right]
$$

$$
\overset{(c)}{\leq} \eta^2 Q \sum_{q'=0}^{q-1} \mathbb{E}^t \left[ \left\| \nabla \mathcal{L}(\mathbf{W}_m^{t,q'}(\mathcal{S}^t), \mathcal{S}^t) - \nabla \mathcal{L}(\mathbf{W}_m^{t,q'}(\mathcal{D}), \mathcal{D}) \right\|^2 \right]
$$

$$
= \eta^2 q \sum_{q'=0}^{q-1} A_m^{t,q'}, \tag{12}
$$

where in step $(a)$ we expand the updates to $\mathbf{W}_m^{t,0}$ with (5) and (6); step $(b)$ cancels $\mathbf{W}_m^{t,0}$ and rearrange the terms; and step $(c)$ applies the Cauchy–Schwarz inequality. At this point, we have the following relations:

$$
\mathbb{E}^t[\|\mathbf{G}^{t,q} - \mathbb{E}^t \, \mathbf{G}^{t,q}\|^2] \leq \sum_{m=1}^{M} A_m^{t,q}, \quad A_0^{t,0} \leq \sigma, \quad A_m^{t,q} \leq 2C_0^2 \eta^2 q \sum_{q'=0}^{q-1} A_m^{t,q'} + 2\sigma, \forall \, q \geq 1.
$$

Note that $q \leq Q$. By choosing $2\eta^2 C_0^2 Q^2 \leq 1$, which implies that $\eta \leq \frac{1}{\sqrt{2}QC_0}$, and by recursively substituting the terms, we have the following bounds:

$$
A_m^{t,q} \leq \left[ 2 + 4q^2\eta^2 C_0^2 + \frac{8}{3}q^3\eta^4 C_0^4 \right] \cdot \sigma \leq \frac{14}{3}\sigma,
$$

$$
\mathbb{E}^t[\|\mathbf{G}^{t,q} - \mathbb{E}^t \, \mathbf{G}^{t,q}\|^2] \leq M \cdot \left[ 2 + 4q^2\eta^2 C_0^2 + \frac{8}{3}q^3\eta^4 C_0^4 \right] \cdot \sigma \leq \frac{14M\sigma}{3}. \tag{13}
$$

This completes bounding the term $\mathbb{E}[\|\mathbf{G}^{t,q} - \mathbb{E}^t \, \mathbf{G}^{t,q}\|^2]$.

### B.2.2 BOUND OF TERM 2

We have the following series of relations:

$$
\mathbb{E}^t \left\| \nabla \mathcal{L}(\mathbf{W}^{t,q}, \mathcal{D}) - \mathbb{E}^t \, \mathbf{G}^{t,q} \right\|^2 = \sum_{m=1}^{M} \mathbb{E}^t \left\| \nabla_{\mathbf{w}_m} \mathcal{L}(\mathbf{W}^{t,q}(\mathcal{S}^t), \mathcal{D}) - \mathbb{E}_\mathcal{S} \, \nabla \mathcal{L}(\mathbf{W}_m^{t,q}(\mathcal{S}), \mathcal{S}) \right\|^2
$$

$$
\overset{(a)}{\leq} \sum_{m=1}^{M} \mathbb{E}^t \, \mathbb{E}_\mathcal{S} \left\| \nabla_{\mathbf{w}_m} \mathcal{L}(\mathbf{W}^{t,q}(\mathcal{S}^t), \mathcal{S}) - \mathbb{E}_\mathcal{S} \, \nabla \mathcal{L}(\mathbf{W}_m^{t,q}(\mathcal{S}), \mathcal{S}) \right\|^2
$$

$$
\overset{(b)}{\leq} \sum_{m=1}^{M} C_0^2 \, \mathbb{E}^t \, \mathbb{E}_\mathcal{S} \left\| \mathbf{W}^{t,q}(\mathcal{S}^t) - [\mathbf{W}_m^{t,q}(\mathcal{S}), \mathbf{W}_{-m}^{t,0}] \right\|^2
$$

$$
= \sum_{m=1}^{M} C_0^2 \, \mathbb{E}^t \, \mathbb{E}_\mathcal{S} \left[ \left\| \mathbf{W}_m^{t,q}(\mathcal{S}^t) - \mathbf{W}_m^{t,q}(\mathcal{S}) \right\|^2 + \sum_{m' \neq m} \left\| \mathbf{W}_{m'}^{t,q}(\mathcal{S}^t) - \mathbf{W}_{m'}^{t,0} \right\|^2 \right]
$$

$$
\overset{(c)}{=} \eta^2 \sum_{m=1}^{M} C_0^2 \, \mathbb{E}^t \, \mathbb{E}_\mathcal{S} \left[ \left\| \sum_{q'=0}^{q-1} \left( \nabla \mathcal{L}(\mathbf{W}_m^{t,q'}(\mathcal{S}^t), \mathcal{S}^t) - \nabla \mathcal{L}(\mathbf{W}_m^{t,q'}(\mathcal{S}), \mathcal{S}) \right) \right\|^2 \right.
$$

$$
\left. + \sum_{m' \neq m} \left\| \sum_{q'=0}^{q-1} \nabla \mathcal{L}(\mathbf{W}_m^{t,q'}(\mathcal{S}); \mathcal{S}) \right\|^2 \right]
$$

$$
\overset{(d)}{\leq} \eta^2 C_0^2 q \sum_{m=1}^{M} \sum_{q'=0}^{q-1} \mathbb{E}^t \, \mathbb{E}_\mathcal{S} \left[ \left\| \nabla \mathcal{L}(\mathbf{W}_m^{t,q'}(\mathcal{S}^t), \mathcal{S}^t) - \nabla \mathcal{L}(\mathbf{W}_m^{t,q'}(\mathcal{S}), \mathcal{S}) \right\|^2 \right.
$$

$$
\left. + \sum_{m' \neq m} \left\| \nabla \mathcal{L}(\mathbf{W}_m^{t,q'}(\mathcal{S}); \mathcal{S}) \right\|^2 \right]
$$

$$\overset{(e)}{=} \eta^2(M+1)C_0^2 q \sum_{m=1}^{M} \sum_{q'=0}^{q-1} \mathbb{E}^t \, \mathbb{E}_{\mathcal{S}} \left\| \nabla \mathcal{L}(\mathbf{W}_m^{t,q'}(\mathcal{S}^t), \mathcal{S}^t) \right\|^2$$

$$\overset{(f)}{=} \eta^2(M+1)C_0^2 q \sum_{q'=0}^{q-1} \mathbb{E}^t \left\| \mathbf{G}^{t,q'} \right\|^2$$

$$= \eta^2(M+1)C_0^2 q \sum_{q'=0}^{q-1} \mathbb{E}^t \left[ \left\| \mathbf{G}^{t,q'} - \mathbb{E}^t \, \mathbf{G}^{t,q'} \right\|^2 + \left\| \mathbb{E}^t \, \mathbf{G}^{t,q'} \right\|^2 \right], \tag{14}$$

where step $(a)$ uses Assumption 3, which states that $\mathcal{S}$ is uniformly sampled from $\mathcal{D}$, and applies Jensen's inequality, that is

$$\left\| \mathbb{E}_{\mathcal{S}} \, \nabla_{\mathbf{W}_m} \mathcal{L}(\mathbf{W}^{t,q}(\mathcal{S}^t); \mathcal{S}) - \mathbb{E}_{\mathcal{S}} \, \nabla \mathcal{L}(\mathbf{W}_m^{t,q}(\mathcal{S}); \mathcal{S}) \right\|^2$$
$$\leq \mathbb{E}_{\mathcal{S}} \left\| \nabla_{\mathbf{W}_m} \mathcal{L}(\mathbf{W}^{t,q}(\mathcal{S}^t); \mathcal{S}) - \nabla \mathcal{L}(\mathbf{W}_m^{t,q}(\mathcal{S}); \mathcal{S}) \right\|^2;$$

step $(b)$ applies Lemma 2 and uses the fact that $\nabla \mathcal{L}(\mathbf{W}_m^{t,q}(\mathcal{S}^t), \mathcal{S}^t)$ is evaluated on $\mathbf{W}_m^{t,q}(\mathcal{S}^t)$ and $\mathbf{W}_{-m}^{t,0}$; in step $(c)$ we expand the update steps until $t, 0$ with (5); step $(d)$ applies Cauchy-Schwarz inequality; in step $(e)$ we reorder the sum and apply the i.i.d. Assumption 3 to $\mathcal{S}, \mathcal{S}^t$; and in step $(g)$ we plug in the definition of $\mathbf{G}$. This completes bounding the term $\mathbb{E}^t \left\| \nabla \mathcal{L}(\mathbf{W}^{t,q}, \mathcal{D}) - \mathbb{E}^t \, \mathbf{G}^{t,q} \right\|^2$.

### B.2.3 Proof of the Main Result

Substituting the last term in (9) with (14), we obtain that the following holds with probability $(1 - \delta)^Q$:

$$\mathbb{E}^t[\mathcal{L}(\mathbf{W}^{t,q+1}, \mathcal{D}) - \mathcal{L}(\mathbf{W}^{t,q}, \mathcal{D})] \leq -\frac{\eta}{2} \mathbb{E}^t \left\| \nabla \mathcal{L}(\mathbf{W}^{t,q}, \mathcal{D}) \right\|^2 - \frac{\eta}{2}(1 - \eta C_0) \left\| \mathbb{E}^t \, \mathbf{G}^{t,q} \right\|^2$$

$$+ \eta^2 C_0 \, \mathbb{E}^t \left\| \mathbf{G}^{t,q} - \mathbb{E}^t \, \mathbf{G}^{t,q} \right\|^2$$

$$+ \frac{1 + \eta C_0}{2C_0} \eta^2 (M+1) C_0^2 q \sum_{q'=0}^{q-1} \mathbb{E}^t \left[ \left\| \mathbf{G}^{t,q'} - \mathbb{E}^t \, \mathbf{G}^{t,q'} \right\|^2 + \left\| \mathbb{E}^t \, \mathbf{G}^{t,q'} \right\|^2 \right]$$

$$\leq -\frac{\eta}{2} \mathbb{E}^t \left\| \nabla \mathcal{L}(\mathbf{W}^{t,q}) \right\|^2 - \frac{\eta}{2}(1 - \eta L) \left\| \mathbb{E}^t \, \mathbf{G}^{t,q} \right\|^2$$

$$+ \frac{1 + \eta C_0}{2C_0} \eta^2 (M+1) C_0^2 q \sum_{q'=0}^{q-1} \mathbb{E}^t \left\| \mathbb{E}^t \, \mathbf{G}^{t,q'} \right\|^2$$

$$+ \eta^2 C_0 \cdot \left( 1 + \frac{(1 + \eta C_0) \cdot (M+1) \cdot \eta Q^2}{2} \right) \cdot \frac{14 M \sigma}{3},$$

where in the second inequality, we set $\eta \leq \frac{1}{\sqrt{2}Q C_0}$, plug in (13), and use the fact that $q \leq Q$. Averaging over $t = 0, \ldots, T - 1$ and $q = 0, \ldots, Q - 1$ and reorganizing the terms, we obtain:

$$\frac{1}{TQ} \sum_{t=0}^{T-1} \sum_{Q=0}^{Q-1} \mathbb{E} \left\| \nabla \mathcal{L}(\mathbf{W}^{t,q}; \mathcal{D}) \right\|^2 \leq \frac{2}{\eta TQ} \mathbb{E}[\mathcal{L}(\mathbf{W}^0) - \mathcal{L}(\mathbf{W}^{T,Q})]$$

$$- \frac{1 - \eta C_0 \left( 1 + (1 + \eta C_0) \cdot (M+1) \cdot Q^2 \right)}{TQ} \sum_{t=0}^{T-1} \sum_{q=0}^{Q-1} \mathbb{E} \left\| \mathbb{E}^t \, \mathbf{G}^{t,q} \right\|^2$$

$$+ 2\eta C_0 \cdot \left( 1 + \frac{(1 + \eta C_0) \cdot (M+1) \cdot \eta Q^2}{2} \right) \cdot \frac{14 M \sigma}{3},$$

which holds with probability at least $(1 - \delta)^{TQ}$. Let $\delta = \delta'/TQ \in (0, 1)$; then, the above equation holds with probability at least

$$(1 - \delta'/TQ)^{TQ} \geq 1 - \delta'/TQ \times TQ = 1 - \delta'.$$

Let

$$1 - \eta C_0 \left( 1 + (1 + \eta C_0) \cdot (M+1) \cdot Q^2 \right) \geq 0,$$

($\eta \leq \frac{1}{\sqrt{M+1}C_0 Q}$) and apply Assumption 2. Then, we have

$$\frac{1}{TQ} \sum_{t=0}^{T-1} \sum_{Q=0}^{Q-1} \mathbb{E} \left\| \nabla \mathcal{L}(\mathbf{W}^{t,q}; \mathcal{D}) \right\|^2 \leq \frac{2(\mathcal{L}(\mathbf{W}^0) - \mathcal{L}^\star)}{\eta TQ} + \frac{28\eta M \cdot (C_0 + \sqrt{M+1}Q)}{3}\sigma, \quad (15)$$

which holds with probability at least $1 - \delta$, where

$$\sigma = 64 G_\ell^2 L_f^2 \log\left(\frac{2dTQ}{\delta}\right) + 128 L_\ell^2 \left(G_f^4 + \frac{1}{S}\right) \left(\log\left(\frac{2dTQ}{\delta}\right) + \frac{1}{4}\right).$$

This completes the proof of Theorem 1.

### B.3    PROOF FOR LEMMA 2

In this subsection, we prove

$$\|\nabla_\mathbf{W} \mathcal{L}(\mathbf{W}) - \nabla_{\mathbf{W}'} \mathcal{L}(\mathbf{W}')\| \leq C_0 \|\mathbf{W} - \mathbf{W}'\|$$

and

$$\left\|\nabla_{\mathbf{W}_m} \mathcal{L}(\mathbf{W}) - \nabla_{\mathbf{W}'_m} \mathcal{L}(\mathbf{W}')\right\| \leq C_0 \|\mathbf{W} - \mathbf{W}'\|.$$

Note that $\nabla_{\mathbf{W}_m} \mathcal{L}(\mathbf{W})$ is a sub-vector of $\nabla \mathcal{L}(\mathbf{W}')$, so $\left\|\nabla_{\mathbf{W}_m} \mathcal{L}(\mathbf{W}) - \nabla_{\mathbf{W}'_m} \mathcal{L}(\mathbf{W}')\right\| \leq \|\nabla_\mathbf{W} \mathcal{L}(\mathbf{W}) - \nabla_{\mathbf{W}'} \mathcal{L}(\mathbf{W}')\|$. Therefore, we only need to prove the first inequality.

The gradient $\nabla \mathcal{L}(\mathbf{W})$ can be expanded as

$$\begin{aligned}
\nabla \mathcal{L}(\mathbf{W}) &= \nabla \ell(\mathbf{y}, f_m(\mathbf{S}, \mathbf{W})) \\
&= \nabla \ell(f_m(\mathbf{S}, \mathbf{W})) \cdot \nabla_\mathbf{W} f_m(\mathbf{S}, \mathbf{W}) = \nabla \ell(f_m) \cdot \nabla f_m(\mathcal{W}),
\end{aligned} \quad (16)$$

where in the last equation we omit the irrelevant variables. Then, we have

$$\begin{aligned}
\|\nabla \mathcal{L}(\mathbf{W}) - \nabla \mathcal{L}(\mathbf{W}')\| &= \|\nabla \ell(f_m) \cdot \nabla f_m(\mathbf{W}) - \nabla \ell(f'_m) \cdot \nabla f_m(\mathbf{W}')\| \\
&= \|\nabla \ell(f_m) \cdot (\nabla f_m(\mathbf{W}) - \nabla f_m(\mathbf{W}')) + (\nabla \ell(f_m) - \nabla \ell(f'_m)) \cdot \nabla f_m(\mathbf{W}')\| \\
&\overset{(a)}{\leq} \|\nabla \ell(f_m) \cdot (\nabla f_m(\mathbf{W}) - \nabla f_m(\mathbf{W}'))\| + \|(\nabla \ell(f_m) - \nabla \ell(f'_m)) \cdot \nabla f_m(\mathbf{W}')\| \\
&\overset{(b)}{\leq} \|\nabla \ell(f_m)\| \|\nabla f_m(\mathbf{W}) - \nabla f_m(\mathbf{W}')\| + \|\nabla \ell(f_m) - \nabla \ell(f'_m)\| \|\nabla f_m(\mathbf{W}')\| \\
&\overset{(c)}{\leq} G_\ell L_f \|\mathbf{W} - \mathbf{W}'\| + L_\ell \|f_m(\mathbf{W}) - f_m(\mathbf{W}')\| \cdot G_f \\
&\overset{A1}{\leq} (G_\ell L_f + L_\ell G_f^2) \cdot \|\mathbf{W} - \mathbf{W}'\|,
\end{aligned} \quad (17)$$

where step $(a)$ uses the fact that $\|a + b\| \leq \|a\| + \|b\|$, step $(b)$ uses the fact that $\|ab\| \leq \|a\| \|b\|$; and in step $(c)$ we apply Lemma 2 (that is, for any $G$-smooth function $g$, its gradient is bounded as $\|\nabla g\| \leq G$) to the first and the fourth terms and Lipschitz gradient to the second and the third terms. This completes the proof of Lemma 2.

## C    EXPERIMENT DETAILS

### C.1    DETAILS OF THE DATASETS

**Planetoid** (Yang et al., 2016): This collection contains three citation datasets: Cora, PubMed, and CiteSeer. Each dataset contains one citation graph, where the nodes represent papers and edges represent citations. The node features are a bag of words and the classification target is the paper category. In the experiment, each client holds a non-overlapping block of node features and a subgraph that results from uniformly sampling 80% of the edges.

**HeriGraph** (Bai et al., 2022): This collection contains three multi-modal graph datasets, each of which is constructed from heritage data posted on social media for a particular city (Suzhou, Amsterdam, and Venice). Each post contains user information, timestamp, geolocation, image, and text annotation. The posts are connected to form three subgraphs: a social subgraph, a spatial subgraph,

and a temporal subgraph. The social subgraph is formed based on friendship and common-interest relations of the users. The spatial subgraph is formed based on the spatial proximity of the geolocations. The temporal subgraph is formed based on the temporal proximity of the posts. Each post has three blocks of image features and possibly text features; for classification, it belongs to one of nine heritage attributes. In the experiment, each client holds one of the three subgraphs and one of the three image feature blocks.

**Reddit** (Hamilton et al., 2017): Reddit is a large online community where users post and comment on different topics. Each node represents a post and the features are the text of the post. Two posts are connected if the same user comments on both. The classification target is the community (subreddit) that a post belongs to. Similar to Planetoid, in the experiment, each client holds a non-overlapping block of node features and a subgraph that results from uniformly sampling 80% of the edges.

## C.2 DETAILS OF THE HYPERPARAMETERS

Here we provide a list of hyperparameters for grid search in Table 5. The optimal set of hyperparameters for each setting is tuned according to the range listed in the table.

| Hyperparameter | Grid search range |
|---|---|
| Hidden dimension of $\mathbf{H}[l]$ | $\{128, 192, 256, 384\}$ |
| Batch size $S$ | $\{16, 32\}$ |
| Neighborhood sample size | $\{2, 3, 4, 6, 8\}$ |
| Training rounds $T$ | $\{512, 640, 1024, 1152, 3200\}$ |
| Learning rate $\eta$ | $\{1, 2, 3.5, 5, 7, 8\} \times \{10^{-1}, 10^{-2}, 10^{-3}\}$ |

Table 5: Hyperparameter grid search range for the numerical experiments.

