# OpenReview forum: "GLASU: A Communication-Efficient Algorithm for Federated Learning with Vertically Distributed Graph Data"
_ICLR.cc/2023/Conference — Submitted to ICLR 2023_

### Official Review · Reviewer_42nY · 2022-10-21

**Confidence:** 3
**Correctness:** 4
**Technical Novelty And Significance:** 2
**Empirical Novelty And Significance:** 2
**Recommendation:** 3

**Clarity, Quality, Novelty And Reproducibility:**

The paper is written clearly. The quality of the experiments can be further improved. The idea of the paper is somehow novel, but not very impressive.

**Strength And Weaknesses:**

Strength: This paper is written clearly and reader-friendly. The mathematical proof of the theorem looks solid.

Weaknesses: The most severe weakness of the paper is that the theorem and experiment did not support the motivation of the algorithm.

    1. Communication-efficient. The motivation of the GLASU algorithm is to save communication. However, neither the theorem nor the experiment discuss the communication cost, which undermines the contribution of the paper.
    2. Privacy is an important aspect in federated learning. But privacy is not discussed in this paper.
    3. The experiment results are not clear enough. For example, in Table 2 and table 4, it is hard to see how Q affects the results. In Table 3, it is hard to see how K affects the results. The experiment results are not strong enough to support the claims of the paper.

**Summary Of The Paper:**

This paper investigates vertical FL on graph neural networks. A communication-efficient vertically-distributed training algorithm called GLASU was proposed, where the main idea for communication saving is lazy aggregation and stale updates. It was proved that the proposed algorithm can converge given that the loss function is smooth with Lipschitz gradient, lower-bounded and the sampling is uniform. Several experiments were conducted.

**Summary Of The Review:**

This paper proposed GLASU, a communication-efficient vFL algorithm on graph neural networks. The convergence of GLASU is proved and some experiments are conducted. However, neither the theorem nor the experiment can prove that the communication cost is reduced. Also, the experiments are not strong enough to support the claim. Hence, the quality of the paper can be further improved.

---

> ### Author Response · Authors · 2022-11-18
> **Response to Reviewer 42nY**
>
> > (1) Communication-efficient. The motivation of the GLASU algorithm is to save communication. However, neither the theorem nor the experiment discuss the communication cost, which undermines the contribution of the paper.
>
> We would like to clarify our theoretical and numerical results.
>
> - On the theoretical side, by using a small stepsize $$\eta = \sqrt{\frac{3(L(W^{0,0})-L^\star)}{14\sigma M(C_0+\sqrt{M+1}Q)QT}} = \mathcal{O}(\frac{1}{Q\sqrt{T}}),$$ the overall convergence rate becomes: $$\frac{1}{TQ}\sum^{T-1,Q-1}_{t,q}\mathbb{E}\|\nabla L(W^{t,q};D)\|^2  \leq  4\sqrt{\frac{14\sigma M(C_0+\sqrt{M+1}Q)(L(W^{0,0})-L^\star)}{QT}}.$$ This suggests that the algorithm converges with rate $\mathcal{O}(\frac{1}{\sqrt{T}}).$ Further, when $C_0\geq\sqrt{M+1}Q$, we have a *linear* speedup in terms of $Q$. That is, to achieve the same $\epsilon$ accuracy (i.e., $\mathbb{E}\|\nabla L(W^{t,q};D)\|^2\leq\epsilon$), with $Q$-step stale updates, it requires $T = \mathcal{O}(\frac{1}{\epsilon^2}\times\frac{1}{Q})$ communications, while *without* stale updates, it requires $T = \mathcal{O}(\frac{1}{\epsilon^2})$ communications.
>
> - On the experiment side, table 4 shows that to achieve the same desired accuracy on the Amsterdam dataset, using more stale updates (from $Q=2$ to $Q=16$) will reduce the communication by $81\\%$. There is also a speedup on the PubMed dataset when $Q$ increases from 2 to 4.
>
> > (2) Privacy is an important aspect in federated learning. But privacy is not discussed in this paper.
>
> We indeed discussed privacy in Sec. 2.5. Specifically, the proposed algorithm is compatible with existing privacy-preserving approaches, including secure aggregation and differential privacy.
>
> > (3) The experiment results are not clear enough. For example, in Table 2 and table 4, it is hard to see how Q affects the results. In Table 3, it is hard to see how K affects the results. The experiment results are not strong enough to support the claims of the paper.
>
> Let us clarify the numerical results.
>
> - In table 2, we compare the proposed algorithm with a few baselines and show that the proposed method can achieve comparable accuracy as centralized models with different choices of $Q$, and it performs significantly better than stand-alone training. The main purpose is to compare across methods, not to show how the performance of our method varies when $Q$ changes.
>
> - To investigate how our performance varies with $Q$, in table 4, we see that when $Q$ is smaller than a certain sweet spot, the communication can be reduced by increasing $Q$. Of course, beyond the sweet spot, using an overly large $Q$ will sacrifice the convergence, which adversely affects the total runtime.
>
> - In table 3, we show how $K$ affects the runtime of the algorithm. It is clearly shown that by using more lazy aggregations (smaller $K$), the runtime significantly drops on both datasets ($37.4\\%$ and $58.2\\%$ decrease, respectively).

---

### Official Review · Reviewer_jr7W · 2022-10-25

**Confidence:** 3
**Correctness:** 3
**Technical Novelty And Significance:** 2
**Empirical Novelty And Significance:** 2
**Recommendation:** 5

**Clarity, Quality, Novelty And Reproducibility:**

The paper is well-organized and easy to follow. But the proposed techniques seem to be applying existing techniques in distributed training to VFL on graph data. Therefore, the novelty is incremental.
No codes are provided, so it could be hard to verify the reproducibility.

**Strength And Weaknesses:**

Strength:
The proposed method reduce the communication time via the proposed communication-reduction technique.

Weaknesses/questions:
1. The model splitting method is very straightforward (including both the averaging and concatenation), and it can hardly be counted as a novel method to me.
2. The proposed lazy aggregation and stale updates are not novel techniques. They have been commonly adopted in distributed training.
3. Is there any further elaboration of the theorem1? How does it compared to centralized training, standalone training, or existing VFL methods? What is the theoretical benefits?
4. 3 clients in HeriGraph is so small. Does the proposed method scale to more clients? What is the client settings in other datasets?
5. What are the real applications of VFL on graph data? Please introduce it along with the applications of horizontal FL in the first graph in related works.
6. Is there any memory concern when "Each client m will store the “all but m” representation" ?

**Summary Of The Paper:**

The author works on the vertical federated learning for graph data. The author proposes a model splitting method to split the model to server and clients, and communication-efficient techniques such as lazy aggregation and stale updates for efficient training. Empirical results show similar performance to centralized training but a smaller wall-clock training time.

**Summary Of The Review:**

The author proposes how to split model to server and clients for VFL on graph data, and proposes lazy aggregation and stale updates for communication efficiency. However, the novelty of those techniques is incremental, given the vast amount of communication-efficient works in distributed training.

---

> ### Author Response · Authors · 2022-11-18
> **Response to Reviewer jr7W (Part I)**
>
> > (1,2) The model splitting method is very straightforward (including both the averaging and concatenation), and it can hardly be counted as a novel method to me. The proposed lazy aggregation and stale updates are not novel techniques. They have been commonly adopted in distributed training.
>
> We would note that the three components (model splitting, stale updates, and lazy aggregation) are organically composed into an effective solution for the problem posed by the paper. The composition itself is novel, even though individual components may not be. The proof that such a composed solution is effective is a challenging problem, requiring theoretical analysis of training convergence as well as empirical evidence of communication saving. Moreover, we note that while model splitting and stale updates are commonly seen, lazy aggregation has not been used in the training of distributed graph neural networks. Existing distributed GNN training algorithms either focus on distributed subgraphs with *full feature* and *partial nodes*, or consider the clients to hold *ego-graphs*. In these cases, lazy aggregation has not been used to the best of our knowledge.
>
> > (3) Is there any further elaboration of the theorem1? How does it compared to centralized training, standalone training, or existing VFL methods? What is the theoretical benefits?
>
> Let us comment on the theoretical benefits. By using a small stepsize $$\eta = \sqrt{\frac{3(L(W^{0,0})-L^\star)}{14\sigma M(C_0+\sqrt{M+1}Q)QT}} = \mathcal{O}(\frac{1}{Q\sqrt{T}}),$$ the overall convergence rate becomes: $$\frac{1}{TQ}\sum^{T-1,Q-1}_{t,q}\mathbb{E}\|\nabla L(W^{t,q};D)\|^2  \leq  4\sqrt{\frac{14\sigma M(C_0+\sqrt{M+1}Q)(L(W^{0,0})-L^\star)}{QT}}.$$ This suggests that the algorithm converges with rate $\mathcal{O}(\frac{1}{\sqrt{T}}).$ Further, when $C_0\geq\sqrt{M+1}Q$, we have a *linear* speedup in terms of $Q$. That is, to achieve the same $\epsilon$ accuracy (i.e., $\mathbb{E}\|\nabla L(W^{t,q};D)\|^2\leq\epsilon$), with $Q$-step stale updates, it requires $T = \mathcal{O}(\frac{1}{\epsilon^2}\times\frac{1}{Q})$ communications, while *without* stale updates, it requires $T = \mathcal{O}(\frac{1}{\epsilon^2})$ communications.
>
> This is the first theoretical analysis for vertical federated learning with GNN. Centralized training, stand-alone training and existing VFL algorithms are not dealing with the same problem, therefore we will be comparing apples with oranges if we are to compare the theoretical results of these algorithms.
>
> > (4) 3 clients in HeriGraph is so small. Does the proposed method scale to more clients? What is the client settings in other datasets?
>
> HeriGraph is a real-world dataset that is by itself distributed in three parts. In practical VFL applications, the number of clients will not be large (mostly less than 10, e.g., [R1] uses 4 clients and [R2] considers 2 clients). The number of clients is generally determined by the construction of data and it cannot be arbitrarily scaled. Moreover, the number of clients is limited by the feature dimension of the nodes, not the graph size.
>
> [R1] Chen, T., Jin, X., Sun, Y., & Yin, W. (2020). Vafl: a method of vertical asynchronous federated learning. _arXiv preprint arXiv:2007.06081_.
>
> [R2] Zhou, J., Chen, C., Zheng, L., Wu, H., Wu, J., Zheng, X., ... & Wang, L. (2020). Vertically federated graph neural network for privacy-preserving node classification. _arXiv preprint arXiv:2005.11903_.

---

> ### Author Response · Authors · 2022-11-18
> **Response to Reviewer jr7W (Part II)**
>
> > (5) What are the real applications of VFL on graph data? Please introduce it along with the applications of horizontal FL in the first graph in related works.
>
> An application of VFL on graph data is to analyze customers' purchase behavior and financial risks. Different graphs of the same set of customers may be constructed by different companies, or even different departments of the same company. Each graph may collect different features of a customer, hence fitting the scenario that we focus on.
>
> > (6) Is there any memory concern when "Each client m will store the “all but m” representation" ?
>
> There is no memory concern. Let us explain in detail:
>
> - When using averaging as the aggregation operation, the "all but m" representation is $H_{-m}[l+1] = H[l+1]- \frac{1}{M}H_m^+[l]$, which has the same size as $H_m[l+1]$. Therefore, the additional memory has size $\sum_{i\in\mathcal{I}}Dim(H_m[l+1])$, which does not increase with $M$ and is smaller than the original model's memory cost.
>
> - When using concatenation as the aggregation operation, the "all but m" representation is $H_{-m}[l+1] = [H_{m'}^{+}[l], m'\neq m].$  Its size linearly scales with $M$. However, we can use a common trick in multi-head attention to remove this scaling. Specifically, we can fix the dimension of $H[l+1]$ as $d_l$ and let the dimension of $H_m[l]$  become $d_l/M$. Then, the additional memory cost is bounded by $\sum_{l\in\mathcal{I}}d_l$, which does not increase with $M$.

---

### Official Review · Reviewer_q1q8 · 2022-10-25

**Confidence:** 3
**Clarity, Quality, Novelty And Reproducibility:** The paper is easy to read. The empiri…
**Correctness:** 3
**Technical Novelty And Significance:** 2
**Empirical Novelty And Significance:** 2
**Recommendation:** 3

**Strength And Weaknesses:**

Issues:
1. The problem set up as it pertains to GNNs and vertical data is not well-motivated.
2. The empirical evaluation is limited to one GNN model, GCNII (simple and deep convolutional network), and 3 clients of FL.
3. The splitting of graph and features is not well explained.

**Summary Of The Paper:**

The paper focuses on learning graph neural networks (GNN) under the federated (FL) scheme. As for FL under vertically distributed data, "each client holds a subgraph of the global graph, part of the features for nodes in this subgraph, and part of the whole model; all clients collaboratively predict node properties." The paper proposes lazy aggregation and stale updates to reduce the communication frequency between clients and the server.

**Summary Of The Review:**

The novelty is lacking and the advantage of the proposed learning scheme is not clear.

Update: I thank the authors' response. I will keep my initial review.

---

> ### Author Response · Authors · 2022-11-18
> **Response to Reviewer q1q8**
>
> > (1,3) The problem set up as it pertains to GNNs and vertical data is not well-motivated. The splitting of graph and features is not well explained.
>
> Let us clarify the problem set and data splitting with the following example: In a use case that analyzes customers' purchase behavior and financial risk, different graphs of the same set of customers may be constructed by different companies, or even different departments of the same company. Each graph may collect different features of a customer, hence the features are vertically split. Each graph is a subgraph of the union of these graphs.
>
> > (2) The empirical evaluation is limited to one GNN model, GCNII (simple and deep convolutional network), and 3 clients of FL.
>
> In fact, we experimented with three different GNN models, including GCN, GCNII, and GAT, in the ablation study. These models are representative in the GNN literature and their unique characteristics (residual connections, pariwise attentions, etc) broadly cover common scenarios met by the training of a graph neural network. The performance of the three models is comparable to the respective performance obtained by centralized training. Such results demonstrate that our proposed algorithm is applicable to a wide range of GNNs.

---

### Official Review · Reviewer_Hn4F · 2022-10-25

**Confidence:** 4
**Clarity, Quality, Novelty And Reproducibility:** It is understandable to a large exten…
**Correctness:** 2
**Technical Novelty And Significance:** 2
**Empirical Novelty And Significance:** 2
**Recommendation:** 3

**Strength And Weaknesses:**

Pros:
1.The paper proposes a communication-efficient GLASU algorithm for federated GNN in a VFL manner.

2. The extensive theoretical analyses are derived to validate the performance guarantees of the proposed method.

3. The algorithm is evaluated on a number of real-world graph datasets to demonstrate the effectiveness of the proposed approach for VFL on graph data.


Cons:
1.	During the training, the clients have to send the aggregated embeddings to the server, which might cause privacy issues, like inference attacks or recovery attacks. It would be better if the author could discuss this in more detail.
2.	Based on my understanding, each client needs to train a local W by minimizing the loss function, which means that each client should hold labels which is a very strong assumption cause in real-world cases, one client usually kept the labels.
3.	The most significant advantage of GNN is that it can aggregate the features through neighbors. However, the proposed method directly concatenates the embeddings of the sub-layer, and the embedding is generated by local clients who keep only partial edges of the whole graph. I do not think the average or concatenation operation can replace or approximate the propagation rule in GNN.
4.	For lazy aggregation, how to choose the indices of K. Furthermore, the embedding of each layer is dependent, and the gradients are calculated by following the chain rules. If skipping one layer, how to keep the consistency of the gradient updates is a big concern.
5.	The paper lacks clarity on how stale updates work and why the proposed method can work without influencing performance.
6.	For Theorem, when T approaches infinite, the gradient norm is still upper-bounded by the second term, and it can not show that the proposed method will finally converge.
7.	The experiments have no running time and accuracy compared with SOTA methods. Moreover, figure 3 shows an unexplainable pattern. It would be better if the author could discuss it in more detail.

**Summary Of The Paper:**

This paper proposed a method of communication-efficient federated learning framework for a vertically distributed graph. Specifically, a lazy aggregation rule is proposed to reduce the communication rounds. A new strategy called stale updates skips aggregation in specific iterations to reduce the cost during vertical training.

**Summary Of The Review:**

In general, the studied problem is interesting and important. In addition, the methodology is principled with three major merits as discussed above. However, the work still has some unaddressed concerns to well justify its technical and empirical contributions.

---

> ### Author Response · Authors · 2022-11-18
> **Response to Reviewer Hn4F (Part I)**
>
> > (1) During the training, the clients have to send the aggregated embeddings to the server, which might cause privacy issues, like inference attacks or recovery attacks. It would be better if the author could discuss this in more detail.
>
> The proposed algorithm is compatible with existing privacy-preserving approaches, including secure aggregation and differential privacy. For example, when the server performs an averaging aggregation, homomorphic encryption can be directly applied. Moreover, differential privacy techniques can be applied either solely or in combination with secure aggregation to our algorithm in server-client communications.
>
> > (2) Based on my understanding, each client needs to train a local W by minimizing the loss function, which means that each client should hold labels which is a very strong assumption cause in real-world cases, one client usually kept the labels.
>
> We thank the reviewer for pointing out this concern. Our method can be extended to accommodate the case where only one client (called A) holds the labels. Here are the changes:
>
> - In the model design, instead of requiring all clients to train the *classifier*, we use only client A to train the classifier.
>
> - In the Sampling step, instead of letting the server sample the training indices $\mathcal{S}[L]$, we use client A to sample $\mathcal{S}[L]$.
>
> - In the JointInference step, client A broadcasts (through the server) the back-propagation loss of the classifier to all the other clients.
>
> > (3) The most significant advantage of GNN is that it can aggregate the features through neighbors. However, the proposed method directly concatenates the embeddings of the sub-layer, and the embedding is generated by local clients who keep only partial edges of the whole graph. I do not think the average or concatenation operation can replace or approximate the propagation rule in GNN.
>
> We thank the reviewer for raising this concern. We address it from three perspectives.
>
> - First, let us clarify that the proposed layer splitting has the ability to *fully recover* the structure of the centralized GNN layer. Recall that $$H_m[l] = H[l], H_m^+[l] = \sigma(A(\mathcal{E_m})H_m[l]W_m[l]), H[l+1] = Agg(H_m^+[l]),$$ and the centralized GNN layer is $$H[l+1] = \sigma(A(\mathcal{E})H[l]W[l]).$$ Then, when the aggregation is a sum, letting $W_m[l] =W[l], \forall m$, and $A(\mathcal{E}) = \sum A(\mathcal{E_m})$, suppose the activation functions are either both activated or deactivated, then the split GCN layer and the centralized GCN layer have the same output.
>
> - Second, as discussed in Sec. 2.4, when $K = L$, the federated GNN model can have *exactly the same* output as the centralized GNN model, as the node embeddings have been propagated over all subgraphs before concatenation/aggregation. When $K<L$, the approximation of the split model to the centralized model becomes coarser, but concatenation still provides the ability to utilize information from other subgraphs. In the experiments, we also empirically show that the split model with different layer clipping can achieve similar performance as the centralized GNN model.
>
> - Lastly, we want to stress that the application of this algorithm is in the case where each client *only* has a subgraph and partial features. It offers a way to perform inter-client collaboration, which will be better than the case where no collaboration occurs.
>
> > (4) For lazy aggregation, how to choose the indices of K. Furthermore, the embedding of each layer is dependent, and the gradients are calculated by following the chain rules. If skipping one layer, how to keep the consistency of the gradient updates is a big concern.
>
> In the experiments, we choose the indices of $K$ evenly across the layers, as intuitively, this helps the clients evenly gather the information from the missing edges. Other choices may be possible, but we do not anticipate that they lead to a noticeable and meaningful difference. For the consistency issue of the gradients, we want to clarify that as each client is training a **partial** model with **different** weights (with possibly different dimensions), the consistency issue does not exist in local training.

---

> ### Author Response · Authors · 2022-11-18
> **Response to Reviewer Hn4F (Part II)**
>
> > (5) The paper lacks clarity on how stale updates work and why the proposed method can work without influencing performance.
>
> Intuitively, the stale updates are performing parallel multi-step block coordinate descent, where each client is updating one block of the whole model in the stale updates. The aggregated inference $\{H^t_{-m}[l+1]\}_{l\in\mathcal{I}}$ stores sufficient (stale) information for client $m$ to perform block descent without synchronization. In theory, we show that such local updates lead to convergence and communication saving.
>
> > (6) For Theorem, when T approaches infinite, the gradient norm is still upper-bounded by the second term, and it can not show that the proposed method will finally converge.
>
> Let us clarify the convergence result: by choosing a small stepsize $$\eta = \sqrt{\frac{3(L(W^{0,0})-L^\star)}{14\sigma M(C_0+\sqrt{M+1}Q)QT}} = \mathcal{O}(\frac{1}{Q\sqrt{T}}),$$ the overall convergence rate becomes: $$\frac{1}{TQ}\sum^{T-1,Q-1}_{t,q}\mathbb{E}\|\nabla L(W^{t,q};D)\|^2  \leq  4\sqrt{\frac{14\sigma M(C_0+\sqrt{M+1}Q)(L(W^{0,0})-L^\star)}{QT}}.$$ This indicates that the algorithm converges with rate $\mathcal{O}(\frac{1}{\sqrt{T}})$.
>
> > (7) The experiments have no running time and accuracy compared with SOTA methods. Moreover, figure 3 shows an unexplainable pattern. It would be better if the author could discuss it in more detail.
>
> As we discussed in Sec. 2.4, the SOTA algorithms (Zhou et al., 2020; Ni et al., 2021) are in fact special cases of our algorithm. In table 3, the case $K=4$ corresponds to Zhou et al. (2020); its runtime is the longest. On the other hand, the case $K=1$ corresponds to Ni et al. (2021); its accuracy is the lowest.
>
> The main message of Figure 3 is that the test performance of different GNN models are comparable. We are unsure what the reviewer means by "unexplainable pattern" but will be happy to clarify further.

---

### Official Review · Reviewer_nqTt · 2022-10-25

**Confidence:** 4
**Correctness:** 3
**Technical Novelty And Significance:** 2
**Empirical Novelty And Significance:** 2
**Recommendation:** 5

**Clarity, Quality, Novelty And Reproducibility:**

### Clarity
* The proposed idea is clearly described, with nice illustrations in Figure 1 and Figure 2.

### Quality
* The technical quality of the proposed GLASU is reasonable, but seems problematic for real-world application. In particular, the idea of sharing layer-level node representations between clients is convincing for GNNs, and the two algorithms (i.e., lazy aggregation and stale updates) to make this idea communication-efficient are convincing as well. However, on the other perspective, the proposed GLASU is designed under the hard-constraint: every client should be in the same stage (See the first weakness above), which limits its real-world application.
* The experimental quality is quite weak, since the authors experiment with the very small number of clients (i.e., 3) for FL, and compare with only one FL baseline. Therefore, more effort is required to show the efficacy of the proposed GLASU.

### Novelty
The novelty is mild, for the following reasons:
* The vertical subgraph FL is not entirely new.
* Also, the idea of sharing layer-level information between client models is not new.
* However, the authors combine and adapt existing schemes well, and also, with subtle node features updating tricks for FL, the authors make the proposed GLASU communication-efficient.


### Reproducibility
* The reproducibility is low, since it is difficult to articulate experimental setups for FL, however, the authors do not provide the source code.

**Strength And Weaknesses:**

### Strengths
* The tackled problem of vertical subgraph FL for graph-structured data is under-explored.
* The proposed subgraph FL scheme with two algorithms for making it communication-efficient is novel, and interesting.
* The authors make effort to theoretically analyze the convergence bound of the proposed GLASU, which is challenging since the estimated stochastic gradient is biased in vertical subgraph FL.

### Weaknesses
* The biggest concern is that every client participating in FL should be in the same stage, to share node representations of specific layers simultaneously between different clients, which is not realistic. Also, considering such the point, it is unclear whether the proposed GLASU, evaluated with 3 clients, is scalable to the larger number of clients (e.g., 10 or 30, which are not large though). I suggest authors to experiment with larger client numbers.
* This paper is not positioned well against existing subgraph FL. In particular, there are several subgraph-level FL [1, 2, 3], and the considered vertical FL scenario is not the only subgraph-level FL. Therefore, I suggest authors to discuss clear differences between the previous subgraph-FL [1, 2, 3], and the targeted vertical subgraph FL.
* The baselines are somewhat weak. The authors compare three baselines, and two (i.e., centralized and local training) of them are not the FL models. Also, the other compared baseline (i.e., Sim) is not directly comparable to the proposed GLASU, since they use different local subgraphs. I am wondering the authors can evaluate discussed methods (Zhou et al., 2020; Liu et al., 2022) by adapting them to the used experimental setups.

---

[1] FedGNN: Federated Graph Neural Network for Privacy-Preserving Recommendation, KDD 2020.

[2] Subgraph Federated Learning with Missing Neighbor Generation, NeurIPS 2020.

[3] Personalized Subgraph Federated Learning, arXiv 2021.

**Summary Of The Paper:**

This paper tackles the vertical Federated Learning (FL) scenario for graph-structured data. In particular, the authors first propose to update the node features calculated from every layer of Graph Neural Network (GNNs), distributed to multiple clients, simultaneously (i.e., at l-th layer, all clients share their node representations from the (l-1)-th layer, and then update the l-th layer node representations). After that, to make such the FL framework communication-efficient, the authors propose not only the lazy aggregation that shares node representations only at the particular layers, but also the stale update that receives node features of other clients from the server at the particular interval. The authors evaluate the proposed GLASU on the vertical subgraph-level FL scenarios, showing the effectiveness and the efficiency of GLASU.

**Summary Of The Review:**

This is a borderline paper: the tackled problem is under-explored and the proposed method has a reasonable design choice; whereas, the experiment is weak, the novelty is mild, and the proposed GLASU might not be applicable to real-world scenarios.

---

> ### Author Response · Authors · 2022-11-18
> **Response to Reviewer nqTt**
>
> > (1) The biggest concern is that every client participating in FL should be in the same stage, to share node representations of specific layers simultaneously between different clients, which is not realistic. Also, considering such the point, it is unclear whether the proposed GLASU, evaluated with 3 clients, is scalable to the larger number of clients (e.g., 10 or 30, which are not large though). I suggest authors to experiment with larger client numbers.
>
> We thank the reviewer for raising the concern. Let us clarify that in the proposed algorithm, only the iterations where communication is performed require synchronization to exchange node representations. In the stale updates, such synchronization is *not* necessary. On the other hand, the assumption that all agents should be at the same stage is a commonly used assumption in the scope of vertical federated learning. For example, in [R1], [R2], the algorithms also require collecting sample information across the nodes, thus requiring all clients to be at the same stage.
>
> Additionally, note that different from distributed training of GNNs, where the graph can be arbitrarily partitioned into subgraphs such that one may desire to see how an algorithm can scale to a large number of partitions, our federated learning case presumes that the graph data by construction is distributed among several clients that cannot be arbitrarily changed or scaled. The number of clients is limited by the feature dimension rather than the graph size, and scaling to a large number of clients is meaningless.
>
> [R1] Gu, B., Xu, A., Huo, Z., Deng, C., & Huang, H. (2021). Privacy-preserving asynchronous vertical federated learning algorithms for multiparty collaborative learning. _IEEE transactions on neural networks and learning systems_.
>
> [R2] Liu, Y., Zhang, X., Kang, Y., Li, L., Chen, T., Hong, M., & Yang, Q. (2022). Fedbcd: A communication-efficient collaborative learning framework for distributed features. _IEEE Transactions on Signal Processing_, _70_, 4277-4290.
>
> > (2) This paper is not positioned well against existing subgraph FL. In particular, there are several subgraph-level FL [1, 2, 3], and the considered vertical FL scenario is not the only subgraph-level FL. Therefore, I suggest authors to discuss clear differences between the previous subgraph-FL [1, 2, 3], and the targeted vertical subgraph FL.
>
> The main difference between our work and these works is that our work deals with *feature-distributed* (vertical) federated learning, while the subgraph-level FL deals with *node-distributed* (horizontal) federated learning. In other words, in our work, the node features are partitioned and the subgraph each client has contains all graph nodes. In subgraph-level FL, the node features are not partitioned but the graph is partitioned. These are fundamentally different settings and admit different use cases.
>
> > (3) The baselines are somewhat weak. The authors compare three baselines, and two (i.e., centralized and local training) of them are not the FL models. Also, the other compared baseline (i.e., Sim) is not directly comparable to the proposed GLASU, since they use different local subgraphs. I am wondering the authors can evaluate discussed methods (Zhou et al., 2020; Liu et al., 2022) by adapting them to the used experimental setups.
>
> We would like to clarify that Zhou et al. (2020), Ni et al. (2021), and Liu et al. (2022) are special cases of our proposed method (as discussed in Sec. 2.4). In the experiments, we did compare with these algorithms. Specifically, the baseline 'Sim' in table 2 is implementing Ni et al. (2021), and case $K=4$ in table 3 is implementing Zhou et al., (2020). The other work Liu et al., (2022) is not used to train GNNs, so we did not compare with it.

---

> > ### Comment · Reviewer_nqTt · 2022-11-26
> > **Further review**
> >
> > Thank you for responding to my initial reviews.
> >
> > ---
> >
> > (1) The biggest concern is that every client participating in FL should be in the same stage, to share node representations of specific layers simultaneously between different clients, which is not realistic.
> >
> > It is unclear why, in the stale updates, such synchronization is not necessary. I still believe the synchronization across all clients is a limitation, and the assumption that all clients should be in the same stage is less convincing.
> >
> > ---
> >
> > (2) This paper is not positioned well against the existing subgraph FL. In particular, there are several subgraph-level FL [1, 2, 3], and the considered vertical FL scenario is not the only subgraph-level FL.
> >
> > I acknowledge that the authors clarify the difference between the existing subgraph FL and the studied vertical graph FL. I suggest authors to further mention various literature on graph FL, such as node, subgraph, and graph-level FL, as well as the studied vertical graph FL, all of which are categorized into graph FL.

---

### Decision · Program_Chairs · 2023-01-20

**Decision:**

Reject

**Justification For Why Not Higher Score:**

Lack of (a) code and experimental details, (b) rigor in theoretical analysis, and (c) effective comparison to baselines.

**Justification For Why Not Lower Score:**

N/A

**Metareview: Summary, Strengths And Weaknesses:**

Paper proposes a new communication efficient algorithm for a specific Vertical FL setting on Graphs using GNNs where features and graph edges are distributed across clients. The algorithm combines (separately known in different settings) techniques like lazy aggregation and stale updates. Authors also provide a theoretical analysis of the algorithm showing convergence. This seems like one of the first papers studies this specific setting. Authors provide experimental comparison to baselines and ablation study. However, the paper lacks full reproduceability due to lack of code and experimental details. For example, it is not clear how run-time was calculated and why run time is not compared with baselines. Overall, it seems like the paper focuses on communication efficiency in VFL, but reviewers brought up that characterization of privacy may be more of a concern there. For example, loss of privacy due to intermediate layer aggregation is not analyzed. Furthermore, theoretical analysis lacks rigor. Objective is not defined properly and analysis doesn't seem to take into account the stale updates and lazy aggregation. Furthermore, the main Lemma 1 proof is omitted even though the concentration analysis seems very involved and subtle due to the complex nature of the algorithm. According to best guess (due to lack of clarity), each of the client's weights may be converging to different stationary points. In that case it is not clear which client's output is used for final prediction.